# Attack-Aware Noise Calibration for Differential Privacy

**Bogdan Kulynych**[*]
Lausanne University Hospital (CHUV)

**Juan Felipe Gomez**[*]
Harvard University

**Georgios Kaissis**
Technical University Munich

**Flavio du Pin Calmon**
Harvard University

**Carmela Troncoso**
EPFL

## Abstract

Differential privacy (DP) is a widely used approach for mitigating privacy risks when training machine learning models on sensitive data. DP mechanisms add noise during training to limit the risk of information leakage. The scale of the added noise is critical, as it determines the trade-off between privacy and utility. The standard practice is to select the noise scale to satisfy a given *privacy budget* $\varepsilon$. This privacy budget is in turn interpreted in terms of operational *attack risks*, such as accuracy, sensitivity, and specificity of inference attacks aimed to recover information about the training data records. We show that first calibrating the noise scale to a privacy budget $\varepsilon$, and then translating $\varepsilon$ to attack risk leads to overly conservative risk assessments and unnecessarily low utility. Instead, we propose methods to directly calibrate the noise scale to a desired attack risk level, bypassing the step of choosing $\varepsilon$. For a given notion of attack risk, our approach significantly decreases noise scale, leading to increased utility at the same level of privacy. We empirically demonstrate that calibrating noise to attack sensitivity/specificity, rather than $\varepsilon$, when training privacy-preserving ML models substantially improves model accuracy for the same risk level. Our work provides a principled and practical way to improve the utility of privacy-preserving ML without compromising on privacy.

## 1   Introduction

Machine learning and statistical models can leak information about individuals in their training data, which can be recovered by membership inference, attribute inference, and reconstruction attacks (Fredrikson et al., 2015; Shokri et al., 2017; Yeom et al., 2018; Balle et al., 2022). The most common defenses against these attacks are based on differential privacy (DP) (Dwork et al., 2014). Differential privacy introduces noise to either the data, the training algorithm, or the model parameters (Chaudhuri et al., 2011). This noise provably limits the adversary's ability to run successful attacks at the cost of reducing the utility of the model.

In DP, the parameters $\varepsilon$ and $\delta$ control the privacy-utility trade-off. These parameters determine the scale (e.g., variance) of the noise added during training: Smaller values of these parameters correspond to larger noise. Larger noise provides stronger privacy guarantees but reduces the utility of the trained model. Typically, $\delta$ is set to a small fixed value (usually between $10^{-8}$ and $10^{-5}$), leaving $\varepsilon$ as the primary tunable parameter. Without additional analyses, the values of parameters $(\varepsilon, \delta)$ alone do not provide a tangible and intuitive operational notion of privacy risk (Nanayakkara et al., 2023). This begs the question: how should practitioners, regulators, and data subjects decide on acceptable values of $\varepsilon$ and $\delta$ and calibrate the noise scale to achieve a desired level of protection?

---

[*]Contributed equally.

38th Conference on Neural Information Processing Systems (NeurIPS 2024).

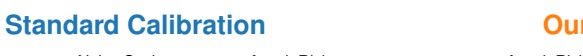

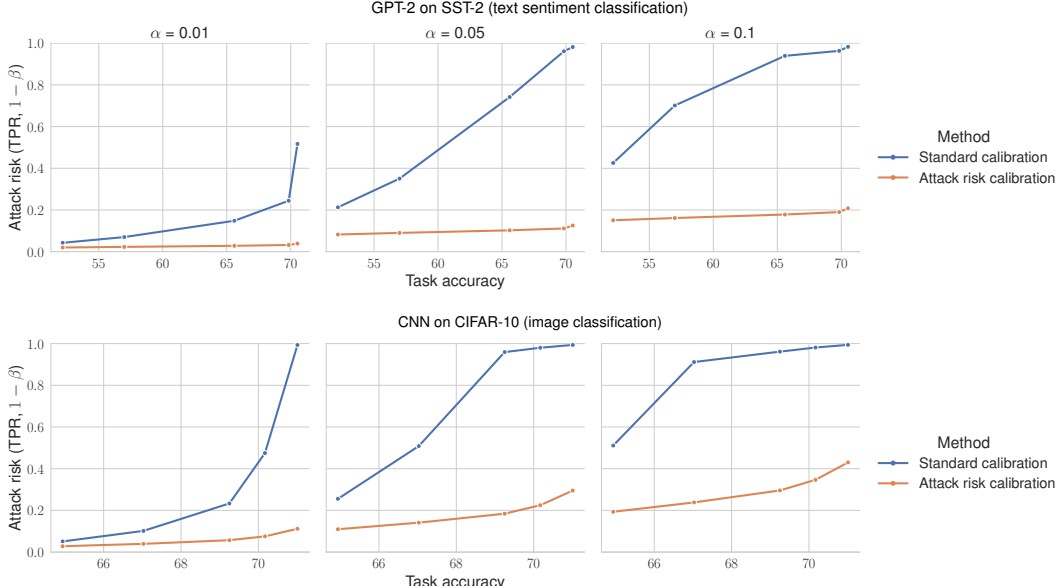

**Figure 1:** Test accuracy (x-axis) of a privately finetuned GPT-2 on SST-2 text sentiment classification dataset (top) and a convolutional neural network on CIFAR-10 image classification dataset (bottom). The DP noise is calibrated to guarantee at most a certain level of privacy attack sensitivity (y-axis) at three possible attack false-positive rates $\alpha \in \{0.01, 0.05, 0.1\}$. See Section 4 for details.

A standard way of assigning operational meaning to DP parameters is mapping them to *attack risks*. One common approach is computing attacker's posterior belief (or equivalently, accuracy or advantage) of membership inference attacks, that concrete values of $(\varepsilon, \delta)$ allow (Wood et al., 2018). An alternative is to compute the trade-off between sensitivity and specificity of feasible membership inference attacks (Wasserman and Zhou, 2010; Kairouz et al., 2015; Dong et al., 2022), which was recently shown to also be directly related to success of record reconstruction attacks (Hayes et al., 2024; Kaissis et al., 2023a). Such approaches map $(\varepsilon, \delta)$ to a quantifiable level of risk for individuals whose data is present in the dataset. Studies have shown that such risk-based measures are the most useful way to interpret the guarantees afforded by DP for practitioners and data subjects (Cummings et al., 2021; Franzen et al., 2022; Nanayakkara et al., 2023).

In this work, we show that directly calibrating the level of noise to satisfy a given level of attack risk, as opposed to satisfying a certain $\varepsilon$, enables a significant increase in utility (see Figure 1). We enable this direct calibration to attack risk by working under $f$-DP (Dong et al., 2022), a hypothesis testing interpretation of DP. In particular, we extend the tight privacy analysis method by Doroshenko et al. (2022) to directly estimate operational privacy risk notions in $f$-DP. Then, we use our extended algorithm to directly calibrate the level of noise to satisfy a given level of attack risk. Concretely, our contributions are:

1. We provide efficient methods for calibrating noise to (a) maximum accuracy (equivalently, advantage), (b) sensitivity and specificity of membership inference attacks, in any DP mechanism, including DP-SGD (Abadi et al., 2016) with arbitrarily many steps.

2. We empirically show that our calibration methods reduce the required noise scale for a given level of privacy risk, up to $2\times$ as compared to standard methods for choosing DP parameters. In a private language modeling task with GPT-2 (Radford et al., 2019), we demonstrate that the decrease in noise can translate to a 18 p.p. gain in classification accuracy.

3. We demonstrate that relying on membership inference accuracy as an interpretation of privacy risk, as is common practice, can increase attack power in privacy-critical regimes, and that calibration for sensitivity and specificity does not suffer from this drawback.

4. We provide a Python package which implements our algorithms for analyzing DP mechanisms in terms of the interpretable $f$-DP guarantees, and calibrating to operational risks:

github.com/Felipe-Gomez/riskcal

Ultimately, we advocate for practitioners to calibrate the noise level in privacy-preserving machine learning algorithms to a sensitivity and specificity constraint under $f$-DP as outlined in Section 3.2.

**Related Work.** Prior work has studied methods for communicating the privacy guarantees afforded by differential privacy (Nanayakkara et al., 2023, 2022; Franzen et al., 2022; Mehner et al., 2021; Wood et al., 2018), and introduced various principled methods for choosing the privacy parameters (Abowd and Schmutte, 2015; Nissim et al., 2014; Hsu et al., 2014). Unlike our approach, these works assume that the mechanisms are calibrated to a given $\varepsilon$ privacy budget parameter, and do not aim to directly set the privacy guarantees in terms of operational notions of privacy risk.

Cherubin et al. (2024); Ghazi and Issa (2023); Izzo et al. (2024); Mahloujifar et al. (2022) use variants of DP that directly limit the advantage of membership inference attacks. We show that calibrating noise to a given level of advantage can increase privacy risk in security-critical regimes and provide methods that mitigate this issue. Leemann et al. (2024) provide methods for evaluating the success of membership inference attacks under a weaker threat model than in DP. Unlike their work, we preserve the standard strong threat model in differential privacy but set and report the privacy guarantees in terms of an operational notion of risk under $f$-DP as opposed to the $\varepsilon$ parameter.

## 2 Problem Statement

### 2.1 Preliminaries

**Setup and notation.** Let $\mathbb{D}^n$ denote the set of all datasets of size $n$ over a space $\mathbb{D}$, and let $S \simeq S'$ denote a neighboring relation, e.g. $S, S'$ that differ by one datapoint. We study randomized algorithms (*mechanisms*) $M(S)$ that take as input a dataset $S \in 2^{\mathbb{D}}$, and output the result of a computation, e.g., statistical queries or an ML model. We denote the output domain of the mechanism by $\Theta$. For ease of presentation, we mainly consider randomized mechanisms that are parameterized by a single noise parameter $\omega \in \Omega$, but our results extend to mechanisms with multiple parameters. For example, in the *Gaussian mechanism* (Dwork et al., 2014), $M(S) = q(S) + Z$, where $Z \sim \mathcal{N}(0, \sigma^2)$ and $q(S)$ is a non-private statistical algorithm, the parameter is $\omega = \sigma$ with $\Omega = \mathbb{R}^{\geq 0}$. We denote a parameterized mechanism by $M_\omega(S)$. We summarize the notation in Table 1 in the Appendix.

**Differential Privacy.** For any $\gamma \geq 0$, we define the hockey-stick divergence from distribution $P$ to $Q$ over a domain $\mathcal{O}$ by

$$D_\gamma(P \parallel Q) \triangleq \sup_E Q(E) - \gamma P(E) \tag{1}$$

where the supremum is taken over all measurable sets $E \subseteq \mathcal{O}$. We define differential privacy (DP) (Dwork et al., 2006) as follows:

**Definition 2.1.** A mechanism $M(\cdot)$ satisfies $(\varepsilon, \delta)$-DP iff $\sup_{S \simeq S'} D_{e^\varepsilon}(M(S) \parallel M(S')) \leq \delta$.

Lower values of $\varepsilon$ and $\delta$ mean more privacy which in turn requires more noise, and vice versa. In the rest of the paper we assume that a larger value of the parameter $\omega \in \Omega$ for $\Omega \subseteq \mathbb{R}$, e.g., standard deviation of Gaussian noise $\omega = \sigma$ in the Gaussian mechanism, means that the mechanism $M_\omega(\cdot)$ is more noisy, which translates into a higher level of privacy (smaller $\varepsilon, \delta$), but lower utility.

Most DP algorithms satisfy a collection of $(\varepsilon, \delta)$-DP guarantees. We define the *privacy profile* (Balle and Wang, 2018), or *privacy curve* (Gopi et al., 2021; Alghamdi et al., 2023) of a mechanism as:

**Definition 2.2.** A parameterized mechanism $M_\omega(\cdot)$ has a privacy profile $\varepsilon_\omega : [0, 1] \to \mathbb{R}$ if for every $\delta \in [0, 1]$, $M_\omega(\cdot)$ is $(\varepsilon(\delta), \delta)$-DP.

We refer to the function $\delta_\omega(\varepsilon)$, defined analogously, also as the privacy profile.

**DP-SGD.** A common algorithm for training neural networks with DP guarantees is DP-SGD (Abadi et al., 2016). The basic building block of DP-SGD is the *subsampled Gaussian mechanism*, defined as $M(S) = q(\mathsf{PoissonSample}_p \circ S) + Z$, where $Z \sim \mathcal{N}(0, \Delta_2^2 \cdot \sigma^2 \cdot I_d)$, and $\mathsf{PoissonSample}_p$ is a procedure which subsamples a dataset $S$ such that every record has the same probability $p \in (0, 1)$ to be in the subsample. DP-SGD, parameterized by $p, \sigma$, and $T \geq 1$, is a repeated application of the

subsampled Gaussian mechanism: $M^{(1)} \circ M^{(2)} \circ \cdots \circ M^{(T)}(S)$, where $q^{(i)}(\cdot)$ is a single step of gradient descent with per-record gradient clipping to $\Delta_2$ Euclidean norm. In line with a standard practice (Ponomareva et al., 2023), we regard all parameters but $\sigma$ as fixed, thus $\omega = \sigma$.

Privacy profiles for mechanisms such as DP-SGD are computed via numerical algorithms called *accountants* (see, e.g., Abadi et al., 2016; Gopi et al., 2021; Doroshenko et al., 2022; Alghamdi et al., 2023). These algorithms compute the achievable privacy profile to accuracy nearly matching the lower bound of a privacy audit where the adversary is free to choose the entire (pathological or realistic) training dataset (Nasr et al., 2021, 2023). Given these results, we regard the analyses of these accountants as tight, and use them for calibration to a particular $(\varepsilon, \delta)$-DP constraint.

**Standard Calibration.** The procedure of choosing the parameter $\omega \in \Omega$ to satisfy a given level of privacy is called *calibration*. In *standard calibration*, one chooses $\omega$ given a target DP guarantee $\varepsilon^\star$ and an accountant that supplies a privacy profile $\varepsilon_\omega(\delta)$ for any noise parameter $\omega \in \Omega$, to ensure that $M_\omega(S)$ satisfies $(\varepsilon^\star, \delta^\star)$-DP:

$$\min_{\omega \in \Omega} \omega \quad \text{s.t. } \varepsilon_\omega(\delta^\star) \geq \varepsilon^\star, \tag{2}$$

with $\delta^\star$ set by convention to $\delta^\star = 1/c \cdot n$, where $n$ is the dataset size, and $c > 1$ (see, e.g., Ponomareva et al., 2023; Near et al., 2023). The parameter $\varepsilon^\star$ is also commonly chosen by convention between 2 and 10 for privacy-perserving ML algorithms with practical utility (Ponomareva et al., 2023). In Eq. (2) and the rest the paper we denote by $\star$ the target value of privacy risk.

After calibration, the $(\varepsilon, \delta)$ parameters are often mapped to some operational notation of privacy attack risk for interpretability. In the next section, we introduce the hypothesis testing framework of DP, $f$-DP, and the notions of risk that $(\varepsilon, \delta)$ parameters are often mapped to. In contrast to standard calibration, in Section 2.3, we calibrate $\omega$ to directly minimize these privacy risks.

## 2.2 Operational Privacy Risks

We can interpret differential privacy through the lens of membership inference attacks (MIAs) in the so-called *strong-adversary model* (see, e.g., Nasr et al., 2021). In this framework, the adversary aims to determine whether a given output $\theta \in \Theta$ came from $M(S)$ or $M(S')$, where $S' = S \cup \{z\}$ for some target example $z \in \mathbb{D}$.[†] The adversary has access to the mechanism $M(\cdot)$, the dataset $S$, and the target example $z \in \mathbb{D}$. Such an attack is equivalent to a binary hypothesis test (Wasserman and Zhou, 2010; Kairouz et al., 2015; Dong et al., 2022):

$$H_0 : \theta \sim M(S), \quad H_1 : \theta \sim M(S'), \tag{3}$$

where the MIA is modelled as a test $\phi : \Theta \to [0, 1]$ that maps a given mechanism output $\theta$ to the probability of the null hypothesis $H_0$ being rejected. We can analyze this hypothesis test through the trade-off between the achievable *false positive rate* (FPR) $\alpha_\phi \triangleq \mathbb{E}_{M(S)}[\phi]$ and *false negative rate* (FNR) $\beta_\phi \triangleq 1 - \mathbb{E}_{M(S')}[\phi]$, where the expectations are taken over the coin flips in the mechanism.[‡] Dong et al. (2022) formalize the *trade-off function* and define $f$-DP as follows:

**Definition 2.3.** A *trade-off function* $T(M(S), M(S')) : [0, 1] \to [0, 1]$ outputs the FNR of the most powerful attack at any given level $\alpha \in [0, 1]$:

$$T(M(S), M(S'))(\alpha) = \inf_{\phi : \Theta \to [0,1]} \{\beta_\phi \mid \alpha_\phi \leq \alpha\} \tag{4}$$

See Figure 5 in the Appendix for an illustration.

**Definition 2.4.** A mechanism $M(\cdot)$ satisfies $f$-DP, where $f$ is the trade-off curve for some other mechanism, if for all $\alpha \in [0, 1]$, we have $\inf_{S \simeq S'} T(M(S), M(S'))(\alpha) \geq f(\alpha)$.

Next, we state the equivalence between $(\varepsilon, \delta)$-DP guarantees and $f$-DP guarantees.

**Proposition 2.1** (Dong et al. (2022)). *If a mechanism $M(\cdot)$ is $(\varepsilon, \delta)$-DP, then it is $f$-DP with*

$$f(\alpha) = \max\{0, \ 1 - \delta - e^\varepsilon \alpha, \ e^{-\varepsilon} \cdot (1 - \delta - \alpha)\}. \tag{5}$$

*Moreover, a mechanism $M(\cdot)$ satisfies $(\varepsilon(\delta), \delta)$-DP for all $\delta \in [0, 1]$ iff it is $f$-DP with*

$$f(\alpha) = \sup_{\delta \in [0,1]} \max\{0, \ 1 - \delta - e^{\varepsilon(\delta)}\alpha, \ e^{-\varepsilon(\delta)} \cdot (1 - \delta - \alpha)\}. \tag{6}$$

---

[†] We use add relation in this exposition, i.e., $S \simeq S'$ iff $S' = S \cup \{z\}$, but our results hold for any relation.
[‡] Note that sensitivity (TPR) is $1 - \beta$ and specificity (TNR) is $1 - \alpha$.

We overview three particular notions of attack risk: advantage/accuracy of MIAs, FPR/FNR of MIAs, and reconstruction robustness. These risks can be thought of as summary statistics of the $f$ curve.

**Advantage/Accuracy.** Wood et al. (2018) proposed[§] to measure the attack risk as the maximum achievable attack accuracy. To avoid confusion with task accuracy, we use *advantage* over random guessing, which is the difference between the attack TPR $1 - \beta_\phi$ and FNR $\alpha_\phi$:

$$\eta \triangleq \sup_{S \simeq S'} \sup_{\phi: \Theta \to [0,1]} 1 - \beta_\phi - \alpha_\phi. \tag{7}$$

The advantage $\eta$ is a linear transformation of the maximum attack accuracy $\sup 1/2 \cdot (1 - \beta_\phi) + 1/2 \cdot (1 - \alpha_\phi)$, where supremum is over $S \simeq S'$ and $\phi : \Theta \to [0,1]$. Moreover, $\eta$ can be obtained from a fixed point $\alpha^* = f(\alpha^*)$ of the $f$ curve as $1 - 2\alpha^*$, and it is bounded given an $(\varepsilon, \delta)$-DP guarantee:

**Proposition 2.2** (Kairouz et al. (2015)). *If a mechanism $M(\cdot)$ is $(\varepsilon, \delta)$-DP, then we have:*

$$\eta \leq \frac{e^\varepsilon - 1 + 2\delta}{e^\varepsilon + 1}. \tag{8}$$

**FPR/FNR Risk.** Recent work (Carlini et al., 2022; Rezaei and Liu, 2021) has argued that MIAs are a relevant threat only when the attack true positive rate $1 - \beta_\phi$ is high at low enough $\alpha_\phi$. As a concrete notion of risk, we thus consider minimum level of attack FNR $\beta^\star$ within an FPR region $\alpha \in [0, \alpha^\star]$, where $\alpha^\star$ is a low value. This approach is similar to the statistically significant p-values often used in the sciences. Following the scientific standards and Carlini et al. (2022), we consider $\alpha^\star \in \{0.01, 0.05, 0.1\}$.

**Reconstruction Robustness.** Another privacy threat is the reconstruction of training data records (see, e.g., Balle et al., 2022). Denoting by $R(\theta; z)$ an attack that aims to reconstruct $z$, its success probability can be formalized as $\rho \triangleq \Pr[\ell(z, R(\theta; z)) \leq \gamma]$ over $\theta \sim M(S \cup \{z\}), z \sim \pi$ for some loss function $\ell : \mathbb{D}^2 \to \mathbb{R}$ and prior $\pi$. Kaissis et al. (2023a) showed that MIA error rates bound reconstruction success as $\rho \leq 1 - f(\kappa_\gamma)$ for an appropriate choice of $\kappa_\gamma$. Therefore, the FPR/FNR trade-off curve can also be thought as a notion of robustness to reconstruction attacks.

## 2.3 Our Objective: Attack-Aware Noise Calibration

The standard practice in DP is to calibrate the noise scale $\omega$ of a mechanism $M_\omega(\cdot)$ to some target $(\varepsilon^\star, \delta^\star)$-DP guarantee, with $\varepsilon^\star$ from a recommended range, e.g., $\varepsilon^\star \in [2, 10]$, and $\delta^\star$ fixed to $\delta^\star < 1/n$, as in Eq. (2). Then, the privacy guarantees provided by the chosen $(\varepsilon^\star, \delta^\star)$ are obtained by mapping these values to bounds on sensitivity and specificity (by Proposition 2.1) or advantage (by Proposition 2.2) of membership inference attacks. In this work, we show that if the goal is to provide an operational and interpretable guarantee such as attack advantage or FPR/FNR, this approach leads to unnecessarily pessimistic noise requirements and a deterioration in utility due to the intermediate step of setting $(\varepsilon^\star, \delta^\star)$. We show it is possible to skip this intermediate step by using the hypothesis-testing interpretation of DP to *directly* calibrate noise to operational notions of privacy risk. In practice, this means replacing the constraint in Eq. (2) with an operational notion of risk:

$$\min_{\omega \in \Omega} \omega \quad \text{s.t. } \texttt{risk}_\omega \leq \texttt{threshold}^\star. \tag{9}$$

Solving this optimization problem requires two components. First, a way to optimize $\omega$ given a method to compute $\texttt{risk}_\omega$. As we assume that risk is monotonic in $\omega$, Eq. (9) can be solved via binary search (see, e.g., Paszke et al., 2019) using calls to the $\texttt{risk}_\omega$ function to an arbitrary precision. Second, we need a way to compute $\texttt{risk}_\omega$ for any value $\omega$. In the next section, we provide efficient methods for doing so for general DP mechanisms, including composed mechanisms such as DP-SGD, by extending the tight privacy analysis from Doroshenko et al. (2022) to computing $f$-DP. Having these methods, we instantiate Eq. (9) for the notions of risks introduced in Section 2.2.

## 3 Numeric Calibration to Attack Risks

In this section, we provide methods for calibrating DP mechanisms to the notions of privacy risk in Section 2.2. As a first step, we introduce the core technical building blocks of our calibration method: methods for evaluating advantage $\eta_\omega$ and the trade-off curve $f_\omega(\alpha)$ for a given value of $\omega$.

---

[§]Wood et al. (2018) used *posterior belief*, which is equivalent to accuracy under uniform prior.

**Dominating Pairs and PLRVs.** We make use of two concepts, originally developed in the context of computing tight privacy profiles under composition: *dominating pairs* (Zhu et al., 2022a) and *privacy loss random variables* (PLRV) (Dwork and Rothblum, 2016).

**Definition 3.1.** We say that a pair of distributions $(P, Q)$ is a *dominating pair* for a mechanism $M(\cdot)$ if for every $\varepsilon \in \mathbb{R}$, we have $\sup_{S \simeq S'} D_{e^\varepsilon}(M(S) \| M(S')) \leq D_{e^\varepsilon}(P \| Q)$.

Importantly, a dominating pair also provides a lower bound on the trade-off curve of a mechanism:

**Proposition 3.1.** *If $(P, Q)$ is a dominating pair for a mechanism $M$, then for $\alpha \in [0, 1]$,*

$$\inf_{S \simeq S'} T(M(S), M(S'))(\alpha) \geq T(P, Q)(\alpha). \tag{10}$$

The proofs of this and all the following statements are in Appendix E. Proposition 3.1 implies that a mechanism $M(\cdot)$ is $f$-DP with $f = T(P, Q)$. Next, we introduce privacy loss random variables, which provide a natural parameterization of the curve $T(P, Q)$.

**Definition 3.2.** Suppose that a mechanism $M(\cdot)$ has a discrete-valued dominating pair $(P, Q)$. Then, we define the *privacy loss random variables* (PLRVs) $(X, Y)$ as $Y \triangleq \log \frac{Q(o)}{P(o)}$, with $o \sim Q$, and $X \triangleq \log \frac{Q(o')}{P(o')}$ with $o' \sim P$.

We can now state the result which serves as a main building block for our calibration algorithms, and forms the main theoretical contribution of our work.

**Theorem 3.3** (Accounting for advantage and $f$-DP with PLRVs)**.** *Suppose that a mechanism $M(\cdot)$ has a discrete-valued dominating pair $(P, Q)$ with associated PLRVs $(X, Y)$. The attack advantage $\eta$ for this mechanism is bounded:*

$$\eta \leq \Pr[Y > 0] - \Pr[X > 0]. \tag{11}$$

*Moreover, for any $\tau \in \mathbb{R} \cup \{\infty, -\infty\}$ and $\gamma \in [0, 1]$, define*

$$\beta^*(\tau, \gamma) = \Pr[Y \leq \tau] - \gamma \Pr[Y = \tau]. \tag{12}$$

*For any level $\alpha \in [0, 1]$, choosing $\tau = (1 - \alpha)$-quantile of $X$ and $\gamma = \frac{\alpha - \Pr[X > \tau]}{\Pr[X = \tau]}$ guarantees that $T(P, Q)(\alpha) = \beta^*(\tau, \gamma)$.*

To show this, we use the Neyman-Pearson lemma to explicitly parameterize the most powerful attack at level $\alpha$ in terms the threshold $\tau$ on the Neyman-Pearson test statistic and the probability $\gamma$ of guessing when the test statistic exactly equals the threshold. See Appendix E.2 for the detailed proof.

We remark that similar results for the trade-off curve appear in (Zhu et al., 2022a) without the $\gamma$ terms, as Zhu et al. assume continuous PLRVs $(X, Y)$. In our work, we rely on the technique due to Doroshenko et al. (2022), summarized in Appendix D, which *discretizes* continuous mechanisms such as the subsampled Gaussian in DP-SGD, and provides a dominating pair that is *discrete* and finitely supported over an evenly spaced grid. As the dominating pairs are discrete, the $\gamma$ terms are non-zero, thus are necessary to fully reconstruct the trade-off curve.

## 3.1 Calibration to Advantage

First, we show how to instantiate Eq. (9) to calibrate noise to a target advantage $\eta^\star \in [0, 1]$. Let $\eta_\omega$ denote the advantage of the mechanism $M_\omega(\cdot)$ as defined in Eq. (7):

$$\min_{\omega \in \Omega} \omega \quad \text{s.t.} \quad \eta_\omega \leq \eta^\star. \tag{13}$$

Given the PLRVs $(X_\omega, Y_\omega)$, we can obtain a substantially tighter bound than converting $(\varepsilon, \delta)$ guarantees using Proposition 2.2 under standard calibration. Specifically, Theorem 3.3 provides the following way to solve the problem:

$$\min_{\omega \in \Omega} \omega \quad \text{s.t.} \quad \Pr[Y_\omega > 0] - \Pr[X_\omega > 0] \leq \eta^\star \tag{14}$$

We call this approach *advantage calibration*, and show how to practically implement it in Algorithms 3 and 4 in the Appendix. Given a method for obtaining valid PLRVs $X_\omega, Y_\omega$ for any $\omega$, such as the one by Doroshenko et al. (2022), advantage calibration is *guaranteed* to ensure bounded advantage, which follows by combining Proposition 3.1 and Theorem 3.3:

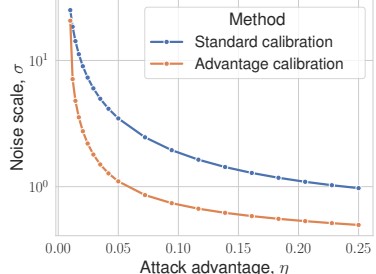
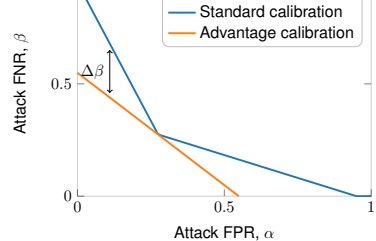

**(a)** Calibrating noise to attack advantage significantly reduces the required noise scale compared to the standard approach. y axis is logarithmic.

**(b)** Optimal calibration for advantage comes with a pitfall: it allows for $\Delta\beta$ higher attack power in the low FPR regime compared to standard calibration.

**Figure 2:** Benefits and pitfalls of advantage calibration.

**Proposition 3.2.** *Given PLRVs $(X_\omega, Y_\omega)$ of a discrete-valued dominating pair of a mechanism $M_\omega(\cdot)$, choosing $\omega^*$ using Eq. (14) ensures $\eta_{\omega^*} \leq \eta^\star$.*

**Utility Benefits.** We demonstrate how calibration for a given level of attack advantage can increase utility. As a mechanism to calibrate, we consider DP-SGD with $p = 0.001$ subsampling rate, $T = 10,000$ iterations, and assume that $\delta^\star = 10^{-5}$. Our goal is to compare the noise scale $\sigma$ obtained via advantage calibration to the standard approach.

As a baseline, we choose $\sigma$ using standard calibration in Eq. (2), and convert the resulting $(\varepsilon, \delta)$ guarantees to advantage using Proposition 2.2. We detail this procedure in Algorithm 2 in the Appendix. We consider target values of advantage $\eta^\star \in [0.01, 0.25]$. As we show in Figure 2a, our direct calibration procedure enables to reduce the noise scale by up to $3.5\times$.

**Pitfalls of Calibrating for Advantage.** Calibration to a given level of membership advantage is a compelling idea due to the decrease in noise required to achieve better utility at the same level of risk as with the standard approach. Despite this increase in utility, we caution that this approach comes with a deterioration of privacy guarantees other than maximum advantage compared to standard calibration. Concretely, it allows for *increased attack TPR* in the privacy-critical regime of low attack FPR (see Section 2.2). The next result quantifies this pitfall:

**Proposition 3.3** (Cost of advantage calibration). *Fix a dataset size $n > 1$, and a target level of attack advantage $\eta^\star \in (\delta^\star, 1)$, where $\delta^\star = 1/c \cdot n$ for some $c > 1$. For any $0 < \alpha < \frac{1-\eta^\star}{2}$, there exists a DP mechanism for which the gap in FNR $f_{\mathsf{standard}}(\alpha)$ obtained with standard calibration for $\varepsilon^\star$ that ensures $\eta \leq \eta^\star$, and FNR $f_{\mathsf{adv}}(\alpha)$ obtained with advantage calibration is lower bounded:*

$$\Delta\beta(\alpha) \triangleq f_{\mathsf{standard}}(\alpha) - f_{\mathsf{adv}}(\alpha) \geq \eta^\star - \delta^\star + 2\alpha\frac{\eta^\star}{\eta^\star - 1}. \tag{15}$$

For example, if we aim to calibrate a mechanism to at most $\eta^\star = 0.5$ (or, 75% attack accuracy), we could potentially increase attack sensitivity by $\Delta\beta(\alpha) \approx 30$ p.p. at FPR $\alpha = 0.1$ compared to standard calibration with $\delta^\star = 10^{-5}$ (see the illustration in Figure 2b). Note that the difference $\Delta\beta$ in Proposition 3.3 is an overestimate in practice: the increase in attack sensitivity can be significantly lower for mechanisms such as the Gaussian mechanism (see Figure 6 in the Appendix).

## 3.2 Safer Choice: Calibration to FNR within a Given FPR Region

In this section, we show how to calibrate the noise in any practical DP mechanism to a given minimum level of attack FNR $\beta^\star$ within an FPR region $\alpha \in [0, \alpha^\star]$, which enables to avoid the pitfalls of advantage calibration. We base this notion of risk off the previous work (Carlini et al., 2022; Rezaei and Liu, 2021) which argued that MIAs are a relevant threat only when the achievable TPR $1 - \beta$ is high at low FPR $\alpha$. We instantiate the calibration problem in Eq. (9) as follows, assuming $M_\omega(\cdot)$ satisfies $f_\omega(\alpha)$-DP:

$$\min_{\omega \in \Omega} \omega \text{ s.t. } \inf_{0 \leq \alpha \leq \alpha^\star} f_\omega(\alpha) \geq \beta^\star. \tag{16}$$

To solve Eq. (16), we begin by showing that such calibration is in fact equivalent to requiring a given level of attack FNR $\beta^\star$ and FPR $\alpha^\star$.

---

**Algorithm 1** Construct the trade-off curve using discrete privacy loss random variables $(X, Y)$

---

**Require:** PMF $\Pr[X_\omega = x_i]$ over grid $\{x_1, x_2, \ldots, x_k\}$ with $x_1 < x_2 < \ldots < x_k$
**Require:** PMF $\Pr[Y_\omega = y_j]$ over grid $\{y_1, y_2, \ldots, y_l\}$ with $y_1 < y_2 < \ldots < y_l$
  1: **procedure** COMPUTEBETA($\omega; \alpha^\star; X_\omega, Y_\omega$)
  2:      $t \leftarrow \min\{i \in \{0, 1, \ldots, k\} \mid \Pr[X_\omega > x_i] \leq \alpha^\star\}$, where $x_0 \triangleq -\infty$
  3:      $\gamma \leftarrow \frac{\alpha^\star - \Pr[X_\omega > x_t]}{\Pr[X_\omega = x_t]}$
  4:      **return** $f_\omega(\alpha^\star) = \Pr[Y_\omega \leq x_t] - \gamma \Pr[Y_\omega = x_t]$

---

**Proposition 3.4.** *For any $\alpha^\star \geq 0, \beta^\star \geq 0$ such that $\alpha^\star + \beta^\star \leq 1$, and any $f$-DP mechanism $M(\cdot)$:*

$$\inf_{0 \leq \alpha \leq \alpha^\star} f(\alpha) \geq \beta^\star \text{ iff } f(\alpha^\star) \geq \beta^\star. \tag{17}$$

This follows directly by monotonicity of the trade-off function $f$ (Dong et al., 2022). The optimization problem becomes:

$$\min_{\omega \in \Omega} \omega \text{ s.t. } f_\omega(\alpha^\star) \geq \beta^\star. \tag{18}$$

Unlike advantage calibration to $\eta^\star$, the approach in Eq. (18) limits the adversary's capabilities without increasing the risk in the privacy-critical low-FPR regime, as we can explicitly control the acceptable attack sensitivity for a given low FPR.

To obtain $f_\omega(\alpha)$, we use the PLRVs $X_\omega, Y_\omega$ along with Theorem 3.3 to compute $f = T(P, Q)$[¶] (see Algorithm 1), and solve Eq. (18) using binary search over $\omega \in \Omega$. We provide the precise procedure in Algorithm 6 in the Appendix. This approach *guarantees* the desired level of risk:

**Proposition 3.5.** *Given PLRVs $(X_\omega, Y_\omega)$ of a discrete-valued dominating pair of a mechanism $M_\omega(\cdot)$, choosing $\omega^*$ using Eq. (18) and Algorithm 1 to compute $f_\omega(\alpha)$ ensures $f_{\omega^*}(\alpha^\star) \geq \beta^\star$.*

### 3.3 Other Approaches to Trade-Off Curve Accounting

In this section, we first contextualize the proposed method within existing work. Then, we discuss settings in which alternatives to PLRV-based procedures could be more suitable.

**Benefits of PLRV-based Trade-Off Curve Accounting.** Computational efficiency is important when estimating $f_\omega(\alpha)$, as the calibration problem requires evaluating this function multiple times for different values of $\omega$ as part of binary search. Algorithm 1 computes $f_\omega(\alpha)$ for a single $\omega$ in $\approx 500$ms, enabling fast calibration, e.g., in $\approx 1$ minute for DP-SGD with $T = 10,000$ steps on commodity hardware (see Appendix H). Existing methods for estimating $f_\omega(\alpha)$, on the contrary, either provide weaker guarantees than Proposition 3.5 or are substantially less efficient. In particular, Dong et al. (2022) introduced $\mu$-GDP, an asymptotic expression for $f_\omega(\alpha)$ as $T \to \infty$, that *overestimates* privacy (Gopi et al., 2021), and thus leads to mechanisms that do not satisfy the desired level of attack resilience when calibrating to it. Nasr et al. (2023); Zheng et al. (2020) introduced a discretization-based approach to approximate $f_\omega(\alpha)$ (discussed next) that can be orders of magnitude less efficient than the direct estimation in Algorithm 1, e.g., 1–6 minutes ($\approx 100$–$700\times$ slower) for a single evaluation of $f_\omega(\alpha)$ in the same setting as before, depending on the coarseness of discretization.

**Calibration using Black-Box Accountants.** Most DP mechanisms are accompanied by $(\varepsilon, \delta)$-DP accountants, i.e., methods to compute their privacy profile $\varepsilon_\omega(\delta)$ or $\delta_\omega(\varepsilon)$. Black-box access to these accountants enables to estimate $\eta_\omega$ and $f_\omega(\alpha)$. In particular, Proposition 2.2 tells us that $(0, \delta)$-DP mechanisms bound advantage as $\eta \leq \delta$. Thus, advantage calibration can also be performed with any $\varepsilon_\omega(\delta)$ accountant by calibrating noise to ensure $\varepsilon_\omega(\eta^\star) = 0$. Estimating $f_\omega(\alpha)$, as mentioned previously, is less straightforward. Existing numeric approaches (Nasr et al., 2023; Zheng et al., 2020) are equivalent to approximating Eq. (6) on a discrete grid over $\delta \in \{\delta_1, \ldots, \delta_u\}$. This requires $u$ calls to the accountant $\varepsilon_\omega(\delta)$, thus quickly becomes inefficient for estimating $f_\omega(\alpha)$ to high precision. We provide a detailed discussion of such black-box approaches in Appendix A.

**Calibration of Mechanisms with Known Trade-Off Curves.** An important feature of our calibration methods is that they enable calibration of mechanisms whose privacy profile is unknown

---

[¶]In practice, we need to additionally symmetrize the trade-off curve due to the implementation details of the add/remove neighborhood relation in the Doroshenko et al. (2022) accountant. See Appendix F.

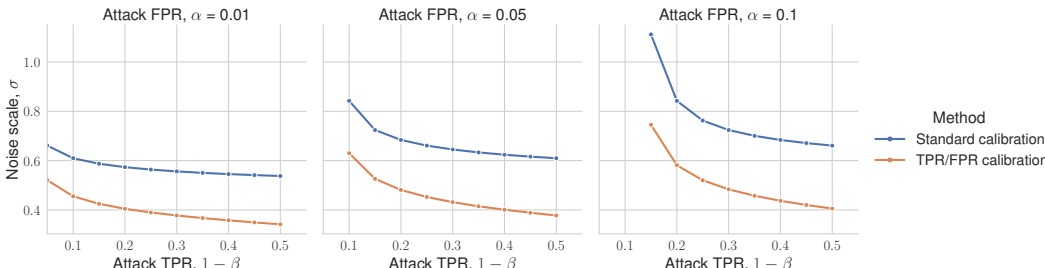

**Figure 3:** Calibration to attack TPR (i.e., $1-$FNR) significantly reduces the noise scale in low FPR regimes. Unlike calibration for attack advantage, this approach does not come with a deterioration of privacy for low FPR, as it directly targets this regime.

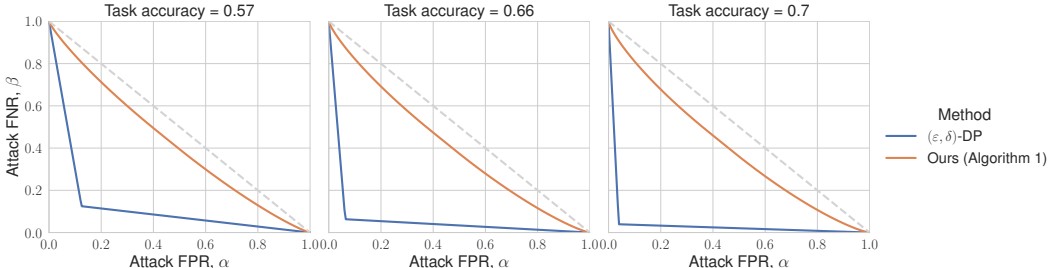

**Figure 4:** Trade-off curves obtained via our method in Algorithm 1 provide a significantly tighter analysis of the attack risks, compared to the standard method of interpreting the privacy risk for a given $(\varepsilon, \delta)$ with fixed $\delta < 1/n$ via Eq. (5). The trade-off curves are shown for three runs of DP-SGD with different noise multipliers in the language modeling experiment with GPT-2. The dotted line - - shows the trade-off curve which corresponds to perfect privacy.

in the exact form, e.g., DP-SGD for $T > 1$. Simpler mechanisms, such as the Gaussian mechanism, which are used for simpler statistical analyses, e.g., private mean estimation, admit exact analytical solutions to the calibration problems in Eqs. (13) and (18). In Appendix G, we provide such solutions for the standard Gaussian mechanism, which enable efficient calibration without needing Algorithm 1.

## 4 Experiments

In this section, we empirically evaluate the utility improvement of our calibration method over traditional approaches. We do so in simulations as well as in realistic applications of DP-SGD. In Appendix H, we also evaluate the utility gain when performing simpler statistical analyses.

**Simulations.** First, we demonstrate the noise reduction when calibrating the DP-SGD algorithm for given error rates using the setup in Section 3.1. We fix three low FPR values: $\alpha^\star \in \{0.01, 0.05, 0.1\}$, and vary maximum attack sensitivity $1 - \beta^\star$ from 0.1 to 0.5 in each FPR regime. We show the results in Figure 3. We observe a significant decrease in the noise scale for all values. Although the decrease is smaller than with calibration for advantage (see Figure 2a), calibrating directly for risk in the low FPR regime avoids the pitfall of advantage calibration: inadvertently increasing risk in this regime.

**Language Modeling and Image Classification.** We showed that FPR/FNR calibration enables to significantly reduce the noise scale. Next, we study how much of this reduction in noise translates into actual utility improvement in downstream applications. We evaluate our method for calibrating noise in private deep learning on two tasks: text sentiment classification using the SST-2 dataset (Socher et al., 2013), and image classification using the CIFAR-10 dataset (Krizhevsky et al., 2009).

For sentiment classification, we fine-tune GPT-2 (small) (Radford et al., 2019) using a DP version of LoRA (Yu et al., 2021). For image classification, we follow the approach of Tramer and Boneh (2021) of training a convolutional neural network on top of ScatterNet features (Oyallon and Mallat, 2015) with DP-SGD (Abadi et al., 2016). See additional details in Appendix H. For each setting, by varying the noise scale, we obtain several models at different levels of privacy. For each of the models

we compute the guarantees in terms of TPR $1 - \beta$ at three fixed levels of FPR $\alpha^\star \in \{0.01, 0.05, 0.1\}$ that would be obtained under standard calibration, and using our Algorithm 1.

Figure 1 shows that FPR/FNR calibration significantly increases *task* accuracy (a notion of utility; not to confuse with *attack* accuracy, a notion of privacy risk) at the same level of $1 - \beta$ for all values of $\alpha^\star$. For instance, for GPT-2, we see the accuracy increase of 18.3 p.p. at the same level of privacy risk (top leftmost plot). To illustrate the reasons behind such a large difference between the methods, in Figure 4, we show the trade-off curves obtained with our Algorithm 1, and with the standard method of deriving the FPR/FNR curve from a single $(\varepsilon, \delta)$ pair for a fixed $\delta < 1/n$ via Eq. (5). We can see that the latter approach drastically overestimates the attack risks, which translates to significantly higher noise and lower task accuracy when calibrating with standard calibration.

## 5 Concluding Remarks

In this work, we proposed novel methods for calibrating noise in differentially private learning targeting a given level of operational privacy risk: advantage and FPR/FNR of membership inference attacks. We introduced an accounting algorithm which directly and tightly estimates privacy guarantees in terms of $f$-DP, which characterizes these operational risks. Using simulations and end-to-end experiments on common use cases, we showed that our attack-aware noise calibration significantly decreases the required level of noise compared to the standard approach at the same level of operational risk. In the case of calibration for advantage, we also showed that the noise decrease could be harmful as it could allow for increased attack success in the low FPR regime compared to the standard approach, whereas calibration for a given level of FPR/FNR mitigates this issue. Next, we discuss limitations and possible directions for future work.

**Choice of Target FPR/FNR.** We leave open the question on how to choose the target FPR $\alpha^\star$ and FNR $\beta^\star$, e.g., whether standard significance levels in sciences such as $\alpha^\star = 0.05$ are compatible with data protection regulation and norms. Further work is needed to develop concrete guidance on the choice of target FPR and FNR informed by legal and practical constraints.

**Catastrophic Failures.** It is possible to construct pathological DP mechanisms which admit catastrophic failures (see, e.g., Ponomareva et al., 2023), i.e., mechanisms which allow non-trivial attack TPR at FPR $\alpha = 0$ so that their trade-off curve is such that $T(M(S), M(S'))(0) < 1$ for some $S \simeq S'$. A classical example in the context of private data release is a mechanism that releases a data record in the clear with probability $\delta > 0$, in which case we have $T(M(S), M(S'))(0) = 1 - \delta$. See the proof of Proposition 3.3 in Appendix E for a concrete construction. In the case that such a pathological mechanism is used in practice, one should use standard calibration to $(\varepsilon, \delta)$ with $\delta \ll 1/n$ to directly limit the chance of catastrophic failures. Fortunately, practical mechanisms such as DP-SGD do not admit catastrophic failures, as they ensure $T(M(S), M(S'))(0) = 1$.

**Tight Bounds for Privacy Auditing.** Multiple prior works on auditing the privacy properties of ML algorithms (Nasr et al., 2021; Liu et al., 2021; Jayaraman and Evans, 2019; Erlingsson et al., 2019) used conversions between $(\varepsilon, \delta)$ and operational risks like in Proposition 2.1, which we have shown to significantly overestimate the actual risks. Beyond calibrating noise, our methods provide bounds on attack success rates for audits in a more precise and computationally efficient way than a recent similar approach by Nasr et al. (2023).

**Accounting in the Relaxed Threat Models.** Although we have focused on DP, our methods apply to any notion of privacy that is also formalized as a hypothesis test. In particular, our method can be used as is to compute privacy guarantees of DP-SGD in a relaxed threat model (RTM) proposed by Kaissis et al. (2023b). Previously, there was no efficient method for accounting in the RTM.

**Applications Beyond Privacy.** Our method can be applied to ensure provable generalization guarantees in deep learning. Indeed, prior work has shown that advantage $\eta$ bounds generalization gaps of ML models (Kulynych et al., 2022a,b). Thus, even though advantage calibration can exacerbate certain risks, it can be a useful tool for ensuring a desired level of generalization in models that usually do not come with non-vacuous generalization guarantees, e.g., deep neural networks.

## Acknowledgements

The authors would like to thank Priyanka Nanayakkara for the helpful suggestions.

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

**Table 1:** Notation summary

| Symbol | Description | Reference |
|---|---|---|
| $z \in \mathbb{D}$ | Data record | |
| $S \in 2^{\mathbb{D}}$ | Dataset of records | |
| $S \simeq S'$ | Adjacency relation of neighboring datasets | |
| $M_\omega : 2^{\mathbb{D}} \to \Theta$ | Privacy-preserving mechanism | |
| $\omega \in \Omega$ | Noise parameter of mechanism $M(S)$ | |
| $D_\gamma(M(S) \parallel M(S')), \gamma \geq 0$ | Hockeystick divergence | Eq. (1) |
| $\varepsilon \in (0, \infty), \delta \in [0, 1]$ | Privacy parameters in differential privacy | Def. 2.1 |
| $\varepsilon_\omega : [0, 1] \to \mathbb{R}$ | Privacy profile curve $\varepsilon_\omega(\delta)$ | Def. 2.2 |
| $\delta_\omega : \mathbb{R} \to [0, 1]$ | Privacy profile curve $\delta_\omega(\varepsilon)$ | Def. 2.2 |
| $\phi : \Theta \to [0, 1]$ | Membership inference hypothesis test | |
| $\alpha_\phi \in [0, 1]$ | False positive rate (FPR) of attack $\phi(\theta)$ | |
| $\beta_\phi \in [0, 1]$ | False negative rate (FNR) of attack $\phi(\theta)$ | |
| $\eta \in [0, 1]$ | Maximal advantage across attacks against mechanism $M(S)$ | Eq. (7) |
| $T(M(S), M(S')) : [0, 1] \to [0, 1]$ | Trade-off curve between FPR and FNR of optimal attacks | Def. 2.3 |
| $f : [0, 1] \to [0, 1]$ | A lower bound on the trade-off curve for all neighboring datasets | Def. 2.4 |
| $P, Q$ | A dominating pair of distributions for a given mechanism $M(S)$ | Def. 3.1 |
| $X, Y$ | Privacy loss random variables for a given dominating pair $P, Q$ | Def. 3.2 |

## A  Attack-Aware Noise Calibration with Black-box DP Accountants

**Advantage Calibration.**  Proposition 2.2 implies that $(0, \delta)$-DP mechanisms ensure bounded advantage $\eta \leq \delta$. Therefore, given access to a black-box accountant $\varepsilon_\omega(\delta)$ or $\delta_\omega(\varepsilon)$ we can calibrate to a given level of advantage $\eta^\star$ by ensuring $(0, \eta^\star)$-DP:

$$\min_{\omega \in \Omega} \omega \quad \text{s.t.} \quad \varepsilon_\omega(\eta^\star) = 0 \quad \text{or} \quad \delta_\omega(0) = \eta^\star \tag{19}$$

This is a more generic way to perform advantage calibration using an arbitrary black-box accountant. It is equivalent to our procedure in Section 3.1 when using Doroshenko et al. (2022) accountant.

**FPR/FNR Calibration with Grid Search.**  Given a black-box DP accountant, i.e., a method which computes the privacy profile $\varepsilon_\omega(\delta)$ of a mechanism $M_\omega(\cdot)$, we can approximate $f_\omega(\alpha)$ by discretizing the range of $\delta \in [0, 1]$ and solving Eq. (6) as:

$$f_\omega(\alpha) \geq \sup_{\delta \in \{\delta_1, \delta_2, \ldots, \delta_u\}} \max\{0, \ 1 - \delta - e^{\varepsilon_\omega(\delta)}\alpha, \ e^{-\varepsilon_\omega(\delta)} \cdot (1 - \delta - \alpha)\}, \tag{20}$$

where $0 \leq \delta_1 < \delta_2 < \ldots < \delta_u \leq 1$. It is possible to perform an analogous discretization using $\delta_\omega(\varepsilon)$ and Proposition 2.1, in which case we have to additionally choose a bounded subspace $\varepsilon \in [\varepsilon_{\min}, \varepsilon_{\max}] \subset \mathbb{R}$. Equivalent procedures to Eq. (20) have previously appeared in Nasr et al. (2023); Zheng et al. (2020).

Plugging in Eq. (20) into the problem in Eq. (18), we can calibrate mechanisms to a given $\alpha^\star, \beta^\star$ using binary search (see Section 2.3) in a space $[\omega_{\min}, \omega_{\max}] \subseteq \Omega$ to additive error $\omega_{\text{err}} > 0$. Denoting by $\nu$:

$$\nu \triangleq \frac{\omega_{\max} - \omega_{\min}}{\omega_{\text{err}}}, \tag{21}$$

the calibration requires $u \cdot \lceil \log_2 \nu \rceil$ evaluations of $\varepsilon_\omega(\delta)$. For instance, a single evaluation of the bound in Eq. (20) takes approximately one minute with $u = 100$, and six minutes with $u = 1,000$ for DP-SGD with $T = 10,000$ using Gopi et al. (2021) accountant as an instantiation of $\varepsilon_\omega(\delta)$ on commodity hardware (see Appendix H). In contrast, evaluating $f_\omega(\cdot)$ using Algorithm 1 in the same settings takes approximately 500ms at the default discretization level $\Delta = 10^{-4}$ (see Appendix D).

Although this approach is substantially less computationally efficient than our direct procedure in Section 3.2, its strength is that it can be used to calibrate noise in any DP algorithm which provides a way to compute its $(\varepsilon, \delta)$ guarantees.

## B  Detailed Calibration Algorithms

**Advantage calibration.**  The standard advantage calibration first finds $\varepsilon^\star$ for a given $\delta^\star < 1/n$ which provides the desired advantage guarantee via Eq. (8), then calibrates noise to the derived $(\varepsilon^\star, \delta^\star)$-DP guarantee using the privacy profile $\varepsilon_\omega(\delta)$ function:

---

**Algorithm 2** Standard advantage calibration

---

**Require:** $\eta^\star, \delta^\star$, where $\delta^\star < \frac{1}{n}$, privacy profile $\varepsilon_\omega(\delta)$.
  1: Find $\varepsilon^\star$ by solving Eq. (8) for $\varepsilon$ with fixed $\delta = \delta^\star$ and $\eta = \eta^\star$
  2: Find noise parameter $\omega^*$, e.g., using binary search:

$$\omega^* \leftarrow \underset{\omega \in \Omega}{\text{argmin}} \text{ s.t. } \varepsilon_\omega(\delta^\star) \geq \varepsilon^\star$$

  3: **return** $\omega^*$

---

For direct calibration to advantage, we first show how to practically use the expression in Theorem 3.3 to evaluate advantage using PLRVs:

---

**Algorithm 3** Compute advantage using PLRVs $(X, Y)$

---

**Require:** PMF $\Pr[X_\omega = \tau]$ over grid $\{x_1, x_2, \ldots, x_k\}$ with $x_1 < x_2 < \ldots < x_k$
**Require:** PMF $\Pr[Y_\omega = \tau]$ over grid $\{y_1, y_2, \ldots, y_l\}$ with $y_1 < y_2 < \ldots < y_l$
  1: **procedure** COMPUTEADV$(\omega; X_\omega, Y_\omega)$
  2:     $t_X \leftarrow \min\{i \in [k] \mid x_i > 0\}$, $t_Y \leftarrow \min\{i \in [l] \mid y_i > 0\}$
  3:     **return** $\sum_{i=t_Y}^{l} \Pr[Y_\omega = y_i] - \sum_{i=t_X}^{k} \Pr[X_\omega = x_i]$

---

Given Algorithm 3, direct calibration to advantage amounts to, e.g., binary search:

---

**Algorithm 4** Direct advantage calibration using PLRVs $(X, Y)$

---

**Require:** $\eta^\star$, PLRVs $X_\omega, Y_\omega$ (see Algorithm 3 for a more detailed specification)
  1: Find noise parameter $\omega^*$, e.g., using binary search:

$$\omega^* \leftarrow \underset{\omega \in \Omega}{\text{argmin}} \text{ s.t. } \text{COMPUTEADV}(\omega; X_\omega, Y_\omega) \leq \eta^\star$$

  2: **return** $\omega^*$

---

**FPR/FNR Calibration.** The standard approach to FPR/FNR calibration proceeds analogously to advantage calibration. First, the algorithm solves Eq. (5) to obtain the value of $\varepsilon^\star$ which ensures that a mechanism satisfies $f(\alpha^\star) = \beta^\star$. Then, the algorithm calibrates the noise to the obtained $(\varepsilon^\star, \delta^\star)$ pair using the privacy profile function $\varepsilon_\omega(\delta)$:

---

**Algorithm 5** Standard FPR/FNR calibration

---

**Require:** $\alpha^\star, \beta^\star, \delta^\star$, where $\delta^\star < \frac{1}{n}$, privacy profile $\varepsilon_\omega(\delta)$.
  1: Find $\varepsilon^\star$ by solving Eq. (5) for $\varepsilon$ with fixed $\delta = \delta^\star$ and $f(\alpha^\star) = \beta^\star$
  2: Find noise parameter $\omega^*$, e.g., using binary search:

$$\omega^* \leftarrow \underset{\omega \in \Omega}{\text{argmin}} \text{ s.t. } \varepsilon_\omega(\delta^\star) \geq \varepsilon^\star$$

  3: **return** $\omega^*$

---

Direct calibration to FPR/FNR amounts to, e.g., binary search, using calls to Algorithm 1:

---

**Algorithm 6** Direct FPR/FNR calibration using PLRVs $(X, Y)$

---

**Require:** $\alpha^\star, \beta^\star$, PLRVs $X_\omega, Y_\omega$ (see Algorithm 1 for a more detailed specification)
  1: Find noise parameter $\omega^*$, e.g., using binary search:

$$\omega^* \leftarrow \underset{\omega \in \Omega}{\text{argmin}} \text{ s.t. } \text{COMPUTEBETA}(\omega; \alpha^\star; X_\omega, Y_\omega) \geq \beta^\star$$

  2: **return** $\omega^*$

---

| Attack risk measure | Symbol | Derived $\beta^\star$ |
|---|---|---|
| Advantage | $\eta^\star$ | $1 - \alpha^\star - \eta^\star$ |
| Accuracy | $\mathrm{acc}^\star$ | $2(\alpha^\star - \mathrm{acc}^\star)$ |
| Positive predictive value / precision | $\mathrm{ppv}^\star$ | $\frac{(\alpha^\star - 1)(\mathrm{ppv}^\star - 1)}{\mathrm{ppv}^\star - 1}$ |

**Table 2:** Some supported risk measures for calibration with a fixed level of FPR $\alpha^\star$, with the derivation of the corresponding level of FNR $\beta^\star$. Given $\alpha^\star$ and the derived $\beta^\star$, we can calibrate noise using the procedure in Section 3.2.

## C  Calibration to Other Risk Notions

Noise calibration for a given FPR/FNR level can be seen as a basic building block to calibrate for other operational measures of risk that are functions of FPR $\alpha$ and FNR $\beta$.

For instance, Rezaei and Liu (2021) propose to measure the risks of membership inference attacks in terms of accuracy $\mathrm{acc}$ and FPR $\alpha$, where: $\mathrm{acc}(\alpha, \beta) \triangleq 1/2 \cdot ((1 - \alpha) + (1 - \beta))$. We can calibrate for a given level of accuracy $\mathrm{acc}^\star$ and FPR $\alpha^\star$ using the method in Section 3.2 by solving the expression for accuracy for a given $\beta^\star$.

Jayaraman et al. (2021) propose to measure positive predictive value, or precision, of attacks:

$$\mathrm{ppv}(\alpha, \beta) \triangleq \frac{1 - \beta}{1 - \beta + \alpha}. \tag{22}$$

Although precision alone is not sufficient to determine the level of privacy, like with accuracy, we can calibrate for a given level of precision $\mathrm{ppv}^\star$ *and FPR* $\alpha^\star$ by deriving the corresponding $\beta^\star$.

We provide the exact conversions in Table 2. These enable practitioners to use the calibration method in Section 3.2 while reporting technically equivalent but potentially more interpretable measures, e.g., attack accuracy at a given FPR.

Although throughout the paper we have assumed that the hypotheses $H_0$ and $H_1$ both have probability $1/2$, our results and conversions can be easily extended to settings where the hypotheses are not equiprobable, as proposed by Jayaraman et al. (2021).

## D  Dominating Pairs

### D.1  Constructing Discrete Dominating Pairs and their PLRVs

We summarize the technique from Doroshenko et al. (2022) to construct a dominating pair from a composed mechanism $M(S) = M^{(1)} \circ M^{(2)} \circ \cdots \circ M^{(T)}(S)$. This models the common use case in privacy-preserving ML where a simple mechanism, such as the subsampled Gaussian in DP-SGD, is applied $T$ times. We assume that each sub-mechanism $M^{(i)}, i \in [T]$, has a known privacy curve $\delta_i(\varepsilon)$. Given an input discretization parameter $\Delta$, a size $k$, and a starting $\varepsilon_1$, (Doroshenko et al., 2022) creates a grid $\{\varepsilon_1, \varepsilon_1 + \Delta, \ldots, \varepsilon_1 + k\Delta\}$. Then, they compute the privacy curve on this grid $\{\delta_i(\varepsilon_1), \delta_i(\varepsilon_1 + \Delta), \ldots, \delta_i(\varepsilon_1 + k\Delta)\}$, and append the values of $\delta(-\infty) = 0$ and $\delta(\infty)$. The dominating pair for the $i^{\text{th}}$ mechanism is constructed using Algorithm 7. Note that Algorithm 7 is identical to Algorithm 1 in Doroshenko et al. (2022), with the notation modified to be consistent with the notation in this paper.

This process is repeated for every mechanism. As long as the discretization parameter $\Delta$ is the same for all $T$ mechanisms, the resulting collection of PLRVs can can be composed via the Fast Fourier Transform. The dominating pair for the composed mechanism $M$ is simply the distribution of $(X_1 + X_2 \ldots + X_T, Y_1 + Y_2 \ldots + X_T)$.

We remark that the discretization parameter $\Delta$ is user-defined, and the choice for the size $k$ and starting $\varepsilon$ for each grid is mechanism-specific. For further implementation details, we point the reader to the code documentation file and the code itself, which can be found in the `dp_accounting` Python library,. In particular, we note that while the PLRVs $X, Y$ have the same support except for atoms at $\pm\infty$, the support of the composed PLRV $X_1 + X_2 \ldots + X_T$ need not be the same as the the support of $Y_1 + Y_2 \ldots + Y_T$. This is because in the convolution part of the implementation of Doroshenko et al. (2022), the code discards any tail probabilities smaller than some truncation

---

**Algorithm 7** (Doroshenko et al., 2022) Construct a dominating pair
___

**Require:** Grid: $\{-\infty, \varepsilon_1, \ldots, \varepsilon_k, \infty\}$.
**Require:** Privacy curve on a grid: $\{0, \delta(\varepsilon_1), \ldots, \delta(\varepsilon_k), \delta(\infty)\}$.
 1: $P(\infty) = 0$
 2: **for** $i = k-1, \ldots, 1$ **do**
 3:      $P(\varepsilon_i) \leftarrow \frac{\delta(\varepsilon_{i-1}) - \delta(\varepsilon_i)}{\exp(\varepsilon_i) - \exp(\varepsilon_{i-1})} - \frac{\delta(\varepsilon_i) - \delta(\varepsilon_{i+1})}{\exp(\varepsilon_{i+1}) - \exp(\varepsilon_i)}$
 4: $P(-\infty) \leftarrow 1 - \sum_{j \in [k-1]} P(\varepsilon_j)$
 5: $Q(-\infty) \leftarrow 0$
 6: **for** $i = 1, \ldots, k-1$ **do**
 7:      $Q(\varepsilon_i) \leftarrow \exp(\varepsilon_i) P(\varepsilon_i)$
 8: $Q(\infty) = \delta(\infty)$
 9: **return** $(P, Q)$
___

parameter. This is why we allow for $X$ and $Y$ to have different support in Algorithm 1, and why we make no assumptions on the distributions of $(P, Q)$ or of $(X, Y)$ in the proof for Theorem 3.3.

## D.2 Some Properties of the Trade-Off Curves of Discrete Dominating Pairs

In this section, we provide several observations on the trade-off curve of discrete dominating pairs. In particular, these observations hold for the trade-off curve described Theorem 3.3.

**Connecting the Dots.** From the proof of Theorem 3.3 (see Appendix E.2), we know that when the level $\alpha$ happens to equal a point in the reverse CDF of $X$, i.e. when $\alpha = \Pr[X > x_i]$ for some $i$, that the corresponding FNR $T(P, Q)(\alpha)$ is simply the CDF of $Y$ evaluated at the same point, i.e. $T(P, Q)(\alpha) = \Pr[Y \leq x_i]$. Since the reverse CDF of $X$ can take on $k+1$ values, it follows that there are $k+1$ values of $\alpha$ where the trade-off curve is fully characterized by the CDF of the PLRVs.

Next, we observe a special structure of the trade-off curve on the points outside of these $k+1$ values. For fixed $\tau$, Eq. (34) implies $\alpha^*(\tau, \gamma)$ is increasing linearly in $\gamma$ and Eq. (37) implies $\beta^*(\tau, \gamma)$ is decreasing linearly in $\gamma$. This implies that the trade-off curve "in between" the $k+1$ points that correspond to the CDFs of the PLRVs is *a linear interpolation*, where one "connects the dots". Hence, the trade-off curve is piece-wise linear, continuous everywhere, and not differentiable at the $k+1$ points where $\alpha$ happens to be on the reverse CDF of $X$.

This observation provides an interesting connection to Doroshenko et al. (2022), who showed that "connecting the dots" between finite points on the *privacy profile* $\delta(e^\varepsilon)$[‖] yields a valid pessimistic estimate to the privacy profile. Could "connecting the dots" in trade-off curve space also yield a valid pessimistic estimate? The answer is clearly no: "connecting the dots" on finite samples from a trade-off curve corresponds to an optimistic bound on the trade-off curve. Nevertheless, it is interesting to note that the class of discrete and finitely supported privacy loss random variables simultaneously achieve a pessimistic bound in privacy profile space and an optimistic bound in trade-off curve space. Further exploration of this phenomena, specifically in the context of constructing optimal optimistic privacy estimates, is left as future work.

**Behavior at the Edges.** The trade-off curve of discrete dominating $(P, Q)$ in general does not satisfy $T(P, Q)(0) = 1$. Indeed, the point $\alpha = 0$ corresponds to $\tau = x_{\max}$ and $\gamma = 0$, in which case $T(P, Q)(0) = \Pr[Y \leq x_{\max}] = 1 - \Pr[Y > x_{\max}]$. Whether or not this equals 1 depends on the details of the PLRV $Y$, though we note that in our experiments, $T(P, Q)(0)$ is usually 1 to within a margin of $10^{-10}$. Moreover, we have that $T(P, Q)(\alpha) = 0$ for any $\alpha \in [\Pr[X > -\infty], 1]$. Indeed, for any $\alpha \in [\Pr[X > -\infty], 1]$, we have that $\tau = -\infty$, meaning that $\beta^*(\tau, \gamma) = \Pr[Y \leq -\infty] = 0$ for any choice of $\gamma$.

The observation that $T(P, Q)(0) \neq 1$, that $T(P, Q)$ is piece-wise linear, and that $T(P, Q)(\alpha) = 0$ for any sufficiently large $\alpha$, are all consistent with the findings of Jin et al. (2023), who characterized the trade-off curves of discrete-valued mechanisms.

___

[‖]The linear interpolation must be done in $e^\varepsilon$ space, as in this grid the privacy profile $\delta(e^\varepsilon)$ is convex.

# E  Omitted Proofs

## E.1  Omitted Proofs in Section 3

First, let us define the notion of the convex conjugate that we use in the proofs. For a given function $f : [0,1] \to [0,1]$, its convex conjugate $f^*$ is:

$$f^*(y) = \sup_{0 \leq x \leq 1} yx - f(x), \tag{23}$$

Next, we can show the omitted proofs.

**Proposition 3.1.** *If $(P,Q)$ is a dominating pair for a mechanism $M$, then for $\alpha \in [0,1]$,*

$$\inf_{S \simeq S'} T(M(S), M(S'))(\alpha) \geq T(P,Q)(\alpha). \tag{10}$$

*Proof.* The proof follows from taking the convex conjugate of both sides of the following result from Zhu et al. (2022b):

**Proposition E.1** (Lemma 20 from Zhu et al. (2022b) restated in our notation). *If a mechanism is $(\varepsilon, D_{e^\varepsilon}(P \parallel Q))$-DP, then it is $f$-DP for $f$ such that the following holds:*

$$D_{e^\varepsilon}(P \parallel Q) = 1 + f^*(-e^\varepsilon).$$

Taking the convex conjugate of the equation above reveals that $f$ follows exactly the structure of the trade-off curve implied by the Neyman-Pearson optimal test, which is exactly $T(P,Q)$. See Appendix E.2.2 for more details on the Neyman-Pearson lemma.

$\square$

**Proposition 3.3** (Cost of advantage calibration). *Fix a dataset size $n > 1$, and a target level of attack advantage $\eta^\star \in (\delta^\star, 1)$, where $\delta^\star = 1/c \cdot n$ for some $c > 1$. For any $0 < \alpha < \frac{1 - \eta^\star}{2}$, there exists a DP mechanism for which the gap in FNR $f_{\mathsf{standard}}(\alpha)$ obtained with standard calibration for $\varepsilon^\star$ that ensures $\eta \leq \eta^\star$, and FNR $f_{\mathsf{adv}}(\alpha)$ obtained with advantage calibration is lower bounded:*

$$\Delta\beta(\alpha) \triangleq f_{\mathsf{standard}}(\alpha) - f_{\mathsf{adv}}(\alpha) \geq \eta^\star - \delta^\star + 2\alpha \frac{\eta^\star}{\eta^\star - 1}. \tag{15}$$

*Proof.* Let us fix a pair of datasets $S \simeq S'$. Suppose that we have a mechanism $M : 2^{\mathbb{D}} \to \{0,1,2,3\}$ which satisfies $(\varepsilon, \delta)$-DP. Further, assume that for the specific fixed pair $S, S'$ it is defined as follows:

$$\begin{array}{ll}
P(M(S) = 0) = 0 & P(M(S') = 0) = \delta \\
P(M(S) = 1) = (1 - \delta) \cdot \frac{e^\epsilon}{e^\epsilon + 1} & P(M(S') = 1) = (1 - \delta) \cdot \frac{1}{e^\epsilon + 1} \\
P(M(S) = 2) = (1 - \delta) \cdot \frac{1}{e^\epsilon + 1} & P(M(S') = 2) = (1 - \delta) \cdot \frac{e^\epsilon}{e^\epsilon + 1} \\
P(M(S) = 3) = \delta & P(M(S') = 3) = 0
\end{array} \tag{24}$$

The defining feature of this mechanism is that its trade-off curve $T(M(S), M(S'))$ for $S, S'$ exactly matches the $f(\cdot)$ curve for generic $(\varepsilon, \delta)$-DP mechanisms in Eq. (5) (Kairouz et al., 2015). Thus, for this mechanism we can use $f$ and $T(M(S), M(S'))$ interchangeably. In the rest of the proof, we assume that we are calibrating this mechanism.

We want to derive (1) $f_{\mathsf{standard}}$ under standard calibration with $\delta^\star = 1/c \cdot n$ and $\varepsilon^\star$ chosen such that we have $\eta \leq \eta^\star$, (2) $f_{\mathsf{adv}}$ under advantage calibration for ensuring $\eta^\star$, and find their difference.

For this, we first solve Eq. (8) for $\varepsilon$ to derive the corresponding $\varepsilon^\star$ that would satisfy the required level of $\eta^\star$ under standard calibration with $\delta^\star = \frac{1}{c \cdot n}$:

$$\varepsilon^\star = \log\left(\frac{2\delta^\star - \eta^\star - 1}{\eta^\star - 1}\right) \tag{25}$$

As we are interested in the low $\alpha$ regime, let us only consider the following form of the DP trade-off curve from Proposition 2.1:

$$f(\alpha) = 1 - \delta - e^\varepsilon \alpha. \tag{26}$$

It is easy to verify that this this form holds for $0 \leq \alpha \leq \frac{1-\delta}{1+e^\varepsilon}$. In the case of $(\varepsilon^\star, \delta^\star)$-DP with $\varepsilon^\star$ defined by Eq. (25), a simple computation shows that this holds for $0 \leq \alpha \leq \frac{1-\eta^\star}{2}$.

To get $f_{\text{standard}}$, we plug $(\varepsilon^\star, \delta^\star)$ into the form in Eq. (26). Recall that by Eq. (8) advantage calibration for generic DP mechanisms is equivalent to calibrating noise to $(0, \eta^\star)$-DP. Thus, to get $f_{\text{adv}}(\alpha)$, we plug into $\varepsilon = 0, \delta = \eta^\star$ to Eq. (26). Subtracting the two, we get:

$$\Delta\beta = \eta^\star - \delta^\star + 2\alpha \frac{\eta^\star - \delta^\star}{\eta^\star - 1}, \tag{27}$$

from which we get the sought form. $\qquad\square$

**Proposition 3.5.** *Given PLRVs $(X_\omega, Y_\omega)$ of a discrete-valued dominating pair of a mechanism $M_\omega(\cdot)$, choosing $\omega^*$ using Eq. (18) and Algorithm 1 to compute $f_\omega(\alpha)$ ensures $f_{\omega^*}(\alpha^\star) \geq \beta^\star$.*

*Proof.* Observe that Algorithm 1 computes the intermediate values of $\tau$ and $\gamma$ considered in the four cases of $\alpha$ values in the proof of Theorem 3.3 given in Appendix E.2, and thus computes the valid trade-off curve $T(P, Q)(\alpha)$ as defined in Eq. (12). By Proposition 3.1, $M_\omega(\cdot)$ satisfies $f$-DP with $f = T(P, Q)$. $\qquad\square$

## E.2  Proof of Theorem 3.3

**Theorem 3.3** (Accounting for advantage and $f$-DP with PLRVs). *Suppose that a mechanism $M(\cdot)$ has a discrete-valued dominating pair $(P, Q)$ with associated PLRVs $(X, Y)$. The attack advantage $\eta$ for this mechanism is bounded:*

$$\eta \leq \Pr[Y > 0] - \Pr[X > 0]. \tag{11}$$

*Moreover, for any $\tau \in \mathbb{R} \cup \{\infty, -\infty\}$ and $\gamma \in [0, 1]$, define*

$$\beta^*(\tau, \gamma) = \Pr[Y \leq \tau] - \gamma \Pr[Y = \tau]. \tag{12}$$

*For any level $\alpha \in [0, 1]$, choosing $\tau = (1 - \alpha)$-quantile of $X$ and $\gamma = \frac{\alpha - \Pr[X > \tau]}{\Pr[X = \tau]}$ guarantees that $T(P, Q)(\alpha) = \beta^*(\tau, \gamma)$.*

Eq. (11) is an implication of a result by Gopi et al. (2021), which states:

$$\delta(\varepsilon) = \Pr[Y > \varepsilon] - e^\varepsilon \Pr[X > \varepsilon]. \tag{28}$$

We get Eq. (11) by observing that $(0, \delta)$-DP bounds $\eta \leq \delta$ from Proposition 2.2.

In the remainder of the proof, we show Eq. (12) and why choosing the threshold $\tau$ and coin flip probability $\gamma$ in the way specified in the theorem guarantees $T(P, Q)(\alpha) = \beta(\tau, \gamma)$. In Appendix E.2.1, we establish the notation necessary for the remainder of the proof along with all the assumptions made. In Appendix E.2.2, we introduce the Neyman-Pearson lemma and use it to construct Eq. (12). Finally, in Appendix E.2.3, we prove the final statement of the theorem.

### E.2.1  Setup, Notation, and Assumptions

Let the domain of $(P, Q)$ be $\mathcal{O}$, which we assume to be countable. We refer to the probability mass function of $P$ as $P(\cdot)$ and similarly for $Q$. We allow for multiple atoms $o$ where $P(o) > 0$ and $Q(o) = 0$, and also multiple atoms $o'$ where $Q(o') > 0$ and $P(o') = 0$. We make no further assumptions on $(P, Q)$.

Since $(P, Q)$ dominate the mechanism $M(\cdot)$, we know from Proposition 3.1 that the hypothesis test:

$$H_0 : o \sim P, \quad H_1 : o \sim Q \tag{29}$$

is easier (the trade-off curve is less than or equal to) that the standard DP hypothesis test:

$$H_0 : \theta \sim M(S), \quad H_1 : \theta \sim M(S') \tag{30}$$

for all $S \simeq S'$. In Appendix E.2.2, we use the Neyman-Pearson Lemma to tightly characterize the trade-off curve implied by (29). The notion of privacy loss random variables (PLRVs) $(X, Y)$, which were defined in Def. 3.2 as $Y \triangleq \log {Q(o)}/{P(o)}$ with $o \sim Q$, and $X \triangleq \log {Q(o')}/{P(o')}$ with $o' \sim P$, appear naturally and play a central role in the proof.

As such, we establish more notation on them. Let $\mathcal{T}$ denote the finite values that the PLRVs can take

$$\mathcal{T} = \{\log Q(o)/P(o) \mid o \in \mathcal{O}, \; P(o) > 0, \; Q(o) > 0\}.$$

We let the support of $X$ be

$$\mathbb{X} = \begin{cases} \{-\infty\} \cup \mathcal{T} & \text{if } \sup \mathcal{T} \in \mathcal{T} \\ \{-\infty\} \cup \mathcal{T} \cup \{\sup \mathcal{T}\} & \text{otherwise.} \end{cases}$$

and we set $\Pr[X = \sup \mathcal{T}] = 0$ if we manually append $\sup \mathcal{T}$ to $X$. We do this to make the quantile of $X$ well-defined on all countable domains. Moreover, let $x_{\max} = \sup \mathbb{X} = \sup \mathcal{T}$. We will often refer to elements in the support of $X$ via $\mathbb{X} = \{-\infty, x_1, x_2, \ldots, x_{\max}\}$.

### E.2.2 Applying the Neyman-Pearson Lemma

According to the Neyman-Pearson Lemma (see, e.g., Lehmann and Romano., 2006; Dong et al., 2022), the most powerful attack at level $\alpha$ for the hypothesis test (29) is a threshold test $\phi^* : \mathcal{O} \to [0, 1]$ parameterized by two numbers $\tau \in \mathbb{R} \cup \{-\infty, \infty\}, \gamma \in [0, 1]$,

$$\phi^*_{\tau,\gamma}(o) = \begin{cases} 1 & \text{if } Q(o) > e^\tau P(o) \\ \gamma & \text{if } Q(o) = e^\tau P(o) \\ 0 & \text{if } Q(o) < e^\tau P(o). \end{cases} \tag{31}$$

which we can equivalently write as:

$$\phi^*_{\tau,\gamma}(o) = \begin{cases} 1 & \text{if } \log \frac{Q(o)}{P(o)} > \tau \\ \gamma & \text{if } \log \frac{Q(o)}{P(o)} = \tau \\ 0 & \text{if } \log \frac{Q(o)}{P(o)} < \tau. \end{cases} \tag{32}$$

This threshold test works by flipping a coin and rejecting the null hypothesis (equivalently, guessing that $o$ came from $Q$) with probability $\phi^*_{\tau,\gamma}(o)$. Here, $\log \frac{Q(o)}{P(o)}$ is the Neyman-Pearson test statistic, and $\tau$ is the threshold for this test statistic. If the test statistic is less (greater) than the threshold, the test always rejects (accepts) the null hypothesis, and if the test statistic equals the threshold, the test flips a coin with probability $\gamma$ to reject the null hypothesis.

The false positive rate of $\phi^*_{\tau,\gamma}$, which we denote by $\alpha$, is the probability that the null hypothesis is rejected ($\phi^*_{\tau,\gamma} > 0$) when the null hypothesis is true ($o \sim P$), and has the following form:

$$\alpha^*(\tau, \gamma) \triangleq \mathop{\mathbb{E}}_{o \sim P}[\phi^*_{\tau,\gamma}(o)] \tag{33}$$

$$= \Pr[X > \tau] + \gamma \Pr[X = \tau]. \tag{34}$$

Similarly, the false negative rate of $\phi^*_{\tau,\gamma}$, which we denote $\beta$, is the probability that the null hypothesis is accepted ($1 - \phi^*_{\tau,\gamma} > 0$) when the null hypothesis is false ($o \sim Q$), and has the following form:

$$\beta^*(\tau, \gamma) \triangleq 1 - \mathop{\mathbb{E}}_{\theta \sim Q}[\phi^*_{\tau,\gamma}(\theta)] \tag{35}$$

$$= 1 - (\Pr[Y > \tau] + \gamma \Pr[Y = \tau]) \tag{36}$$

$$= \Pr[Y \leq \tau] - \gamma \Pr[Y = \tau]. \tag{37}$$

We have thus shown the correctness of the construction of Eq. (12). In Appendix E.2.3, we prove the final statement in Theorem 3.3.

### E.2.3 Construction of the Trade-Off Curve of a Dominating Pair

The goal of this section is to prove the following statement made in Theorem 3.3:

---

For any level $\alpha \in [0, 1]$, choosing $\tau = (1 - \alpha)$-quantile of $X$ and $\gamma = \frac{\alpha - \Pr[X > \tau]}{\Pr[X = \tau]}$ guarantees that:

$$T(P, Q)(\alpha) = \beta^*(\tau, \gamma).$$

---

where $T(P, Q)(\alpha)$ outputs the false negative rate of the most powerful attack at level $\alpha$. From Appendix E.2.2, we know that the most powerful attack takes the form $\phi^*_{\tau,\gamma}$ as defined in Eq. (32).

One should think of the level $\alpha$ as a constraint on the attack $\phi^*_{\tau,\gamma}$. In particular, the constraint $\alpha^*(\tau, \gamma) = \alpha$ (where $\alpha^*$ is the false positive rate of $\phi^*_{\tau,\gamma}$ and is defined in Eq. (33)) yields a family of possible tests that all achieve the level $\alpha$. If $(P, Q)$ were continuous distributions, the constraint $\alpha^*(\tau, \gamma) = \alpha$ would uniquely determine the optimal test. This does not hold in the discrete case, and hence we must identify the most powerful test within this family.

Below, we list out 4 different regimes for the value of the level $\alpha$, identify the family of possible tests in each regime and the most powerful test, and finally give the false negative rate of the respective most powerful test.

1 **Case $\alpha = 1$:** Recall that $X$ has a finite probability of being $-\infty$, meaning that the only way to have $\alpha^*(\tau, \gamma) = 1$ is to set $\tau = -\infty$ and $\gamma = 1$. The corresponding false negative rate is given by $\beta^*(-\infty, 1) = \Pr[Y \le -\infty] - \Pr[Y = -\infty] = 0$.

2 **Case $\alpha = 0$:** If we choose the threshold $\tau = x_{\max}$ and the coin flip probability $\gamma = 0$, then we have that the false positive rate of this test is:

$$\alpha^*(\tau = x_{\max}, \gamma = 0) = \Pr(X > x_{\max}) + \gamma \Pr[X = x_{\max}] \tag{38}$$
$$= 0. \tag{39}$$

Moreover, any test with $\tau > x_{\max}$ has $\alpha^*(\tau, \gamma) = 0$. However, increasing the threshold above $x_{\max}$ can never decrease $\beta^*$. Moreover, a test with a threshold $\tau < x_{\max}$ cannot achieve $\alpha = 0$. It follows that choosing $(\tau = x_{\max}, \gamma = 0)$ yields the most powerful test, which has a false negative rate of $\beta^*(x_{\max}, 0) = \Pr[Y \le x_{\max}]$.

3 **Case $\alpha = \Pr[X > x_t]$ for some $x_t \in \mathbb{X}$:** If we choose the threshold $\tau = x_t$ and coin flip probability $\gamma = 0$, then we have that the false positive rate of this test is

$$\alpha^*(\tau = x_t, \gamma = 0) = \Pr(X > x_t) + 0 \tag{40}$$
$$= \alpha. \tag{41}$$

Moreover, the test $\phi^*_{x_{t+1},1}$ and any test with $\tau \in (x_t, x_{t+1})$ has $\alpha^*(\tau, \gamma) = \alpha$. It is straightforward to see that all these tests are equivalent to outputting 1 if $\log \frac{Q(o)}{P(o)} > x_t$ and 0 otherwise, making them all equivalent to $\phi^*_{x_t,0}$. Note that no other test can achieve the level $\alpha$, since decreasing the threshold below $x_t$ or above $x_{t+1}$ makes it impossible to achieve level $\alpha$. For fixed threshold $\tau = x_t$ $(x_{t+1})$, only a coin flip probability of $\gamma = 0(1)$ achieves level $\alpha$. We conclude that all the tests that achieve level $\alpha$ have a false negative rate of $\beta^* = \Pr[Y \le x_t]$.

4 **Otherwise:** If we choose the threshold

$$\tau = \inf\{x \in \mathbb{X} \mid \alpha \ge \Pr[X > x]\} \tag{42}$$

and choose the coin flip probability $\gamma$ to exactly satisfy the constraint that $\alpha^*(\tau, \gamma) = \alpha$, i.e.,

$$\gamma = \frac{\alpha - \Pr[X > x_t]}{\Pr[X = x_t]}, \tag{43}$$

then this test achieves a false positive rate of $\alpha$. It is easy to see that this is the only test that achieves level $\alpha$, and has a false negative rate of $\beta^* = \Pr[Y \le x_t] - \gamma \Pr[Y = x_t]$.

Note that in all regimes, there is one unique test that achieves a level $\alpha$ and is the most powerful test. However, in some regimes of $\alpha \in [0, 1]$, namely regime 3, there are many *different parameterizations* for the same test. In these cases, we are free to choose any parameterization. For each regime, the very first test we list is the parameterization we choose. To summarize, we have the following most powerful tests:

    1 when $\alpha = 1$, choose $\tau = -\infty, \gamma = 1$

    2 when $\alpha = 0$, choose $\tau = x_{\max}, \gamma = 0$

    3 when $\alpha = \Pr[X > x_t]$, choose $\tau = x_t, \gamma = 0$

    4 else, choose $\tau$ via Eq. (42), and $\gamma = \frac{\alpha - \Pr[X > \tau]}{\Pr[X = \tau]}$.

It is clear from the list above that for distributions with finite support, the most powerful test can be concisely written as:

$$\tau = \inf\{x \in \mathbb{X} \mid \alpha \geq \Pr[X > x]\} \tag{44}$$

$$\gamma = \frac{\alpha - \Pr[X > \tau]}{\Pr[X = \tau]}. \tag{45}$$

where we recognize $\tau$ as the $(1 - \alpha)$-quantile of $X$.

Note that for distributions with countably infinite support, Eq. (45) does not capture Case 2, since $\Pr[X = x_{\max}] = 0$. So, we define $\gamma = 0$ whenever $\alpha = 0$, and $\gamma =$ Eq. (45) otherwise. Since this work focuses on using PLRVs from Doroshenko et al. (2022), which are always finitely supported, we report Eq. (44) and Eq. (45) without this edge case in the main body.

We remark that similar results regarding the trade-off curve between two discrete mechanisms can be found in Jin et al. (2023). We differ from this work by parameterizing the trade-off curve using PLRVs, in contrast to Jin et al., who parameterized the trade-off curve in terms of the discrete distributions $P$ and $Q$. Our parameterization lends itself more naturally to composition, as the PLRVs sum under composition.

# F  Practical Considerations

The algorithm of Doroshenko et al. (2022), which is implemented in the `dp_accounting` Python library,[**] handles Poisson subsampling under composition (i.e. accounting for DP-SGD) by analyzing the removal and add relations separately. This approach, to the authors knowledge, was first advocated for by Zhu et al. (2022b) (see the discussion in their Appendix).

In particular, instead of the algorithm outputting a dominating pair $(P, Q)$ that dominates for the symmetric add/remove relation under composition, it outputs one dominating pair for the asymmetric remove relation $(P_{\text{remove}}, Q_{\text{remove}})$ and one for the asymmetric add relation $(P_{\text{add}}, Q_{\text{add}})$. This means that naively applying Theorem 3.3 to, for example, $(P_{\text{add}}, Q_{\text{add}})$, will return a trade-off curve that is only valid for DP-SGD under the asymmetric add relation.

To handle the case when Theorem 3.3 is applied to a dominating pair $(P, Q)$ (equivalently, the PLRVS $(X, Y)$) that only dominate a mechanism under an asymmetric neighboring relation, a more sophisticated technique is needed to map $T(P, Q)$ to the target symmetric neighboring relation. In particular, a result from (Dong et al., 2022) explains how to handle this case:

**Proposition F.1** (Proposition F.2 from Dong et al. (2022)). *Let $f : [0, 1] \to [0, 1]$ be a convex, continuous, non-increasing function with $f(x) \leq 1 - x$ for $x \in [0, 1]$. Suppose a mechanism $M$ is $(\varepsilon, 1 + f^*(-e^\varepsilon))$-DP for all $\varepsilon \geq 0$, then it is Symm$(f)$-DP with the symmetrization operator Symm$(f)$ defined as:*

$$Symm(f)(x) = \begin{cases} \{f, f^{-1}\}^{**}, & \text{if } \bar{x} \leq f(\bar{x}), \\ \max\{f, f^{-1}\}, & \text{if } \bar{x} > f(\bar{x}), \end{cases} \tag{46}$$

*where $\bar{x} = \inf\{x \in [0, 1] \mid : -1 \in \partial f(x)\}$, and*

$$\{f, f^{-1}\}^{**}(x) = \begin{cases} f(x), & \text{if } x \leq \bar{x}, \\ f(\bar{x}), & \text{if } \bar{x} < x \leq f(\bar{x}), \\ f^{-1}(x), & \text{if } x > f(\bar{x}). \end{cases} \tag{47}$$

Though not explicitly stated, the proposition does assume the mechanism $M(\cdot)$ has a symmetric neighboring relation. By letting $f$ be unspecified however, the proposition allows for the input function $f$ to correspond to an asymmetric neighboring relation. In this case, the proposition returns a trade-off curve that holds for the symmetric neighboring relation.

We can hence apply this proposition to the problem at hand by recalling that given a dominating pair $(P, Q)$, we have that the mechanism is $(\varepsilon, D_{e^\varepsilon}(P \parallel Q))$-DP. Moreover, Theorem 3.3 outputs the trade-off function $f = T(P, Q)$, which is exactly the function $f$ such that $D_{e^\varepsilon}(P \parallel Q) = 1 + f^*(-e^\varepsilon)$. We can thus restate Proposition F.1 in more familiar form as:

---

[**]https://github.com/google/differential-privacy/tree/main/python/dp_accounting/dp_accounting/pld

**Proposition F.2** (Proposition F.2 from Dong et al. (2022) restated). *Suppose that $(P, Q)$ is a dominating pair for a mechanism $M(\cdot)$ under either the add or remove relation. Then, the mechanism is $\mathrm{Symm}(T(P, Q))$-DP with respect to the add/remove relation.*

Proposition F.2 allows us to, for example, use a dominating pair for the asymmetric add relation to obtain a trade-off curve for the symmetric add/remove relation. Moreover, the operator $\mathrm{Symm}(T(P, Q))$ turns out to be straightforward to implement in practice.

Appendix E.2.3 details how to explicitly construct $T(P, Q)$. It is well known that $T(Q, P)(\alpha) = T(P, Q)^{-1}(\alpha)$, hence the order of $(P, Q)$ can be easily swapped in Appendix E.2.3 to get the inverse function $T(P, Q)^{-1}$. The only obstacle remaining is in determining $\bar{x} = \inf\{x \in [0, 1] \mid : -1 \in \partial f(x)\}$. Due to the structure of $T(P, Q)$, namely that it is a piece-wise linear function parameterized by Eq. (34) and Eq. (37), it turns out that the subdifferential $\partial f(x)$ are of the form $\{e^\tau\}$, where $\tau$ are the allowable thresholds of the Neyman-Pearson lemma at level $x$ identified in each of the 4 cases of the proof laid out in Appendix E.2.3. As an example, a unique threshold of $-\infty$ at $\alpha = 1$ implies that the derivative of $T(P, Q)$ at $\alpha = 1$ is 0, meaning the trade-off curve is flat there.

It follows that the constraint $\bar{x} = \inf\{x \in [0, 1] \mid : -1 \in \partial f(x)\}$ implies that $\bar{x}$ is the smallest level $\alpha$ where the threshold switches signs, i.e. $\bar{x} = \alpha^*(\tau = 0, \gamma = 0) = \Pr[X > 0]$ and $f(\bar{x}) = \beta^*(\tau = 0, \gamma = 0) = \Pr[Y \leq 0]$. This gives us all the information needed to implement the Symm operator.

# G   Calibrating Gaussian Mechanism

In the case where the trade-off curve of the mechanism has a closed form, we can solve the calibration problems in Eqs. (13) and (18) exactly without resorting to the numerical procedures in Sections 3.1 and 3.2.

**Definition G.1.** For a given non-private algorithm $q : 2^{\mathbb{D}} \to \mathbb{R}^d$, a Gaussian mechanism (GM) is defined as $M(S) = q(S) + \xi$, where $\xi \sim \mathcal{N}(0, \Delta_2 \cdot \sigma^2 \cdot I_d)$ and $\Delta_2 \triangleq \sup_{S \simeq S'} \|q(S) - q(S')\|_2$ is the *sensitivity* of $q(S)$.

For the Gaussian mechanism, we can exactly compute the relevant adversary's error rates:

**Proposition G.1** (Balle and Wang (2018); Dong et al. (2022)). *Suppose that $M_\sigma(S)$ is GM with sensitivity $\Delta_2$ and noise variance $\sigma^2$. Denote by $\mu = \Delta_2/\sigma$ and by $\Phi(t)$ the CDF of the standard Gaussian distribution $\mathcal{N}(0, 1)$. Then,*

- *The mechanism satisfies $(\varepsilon, \delta)$-DP if the following holds:*

$$\delta = \Phi\left(\frac{\mu}{2} - \frac{\varepsilon}{\mu}\right) - e^\varepsilon \Phi\left(-\frac{\mu}{2} - \frac{\varepsilon}{\mu}\right) \tag{48}$$

- *It satisfies $f$-DP with:*

$$f(\alpha) = \Phi\left(\Phi^{-1}(1 - \alpha) - \mu\right) \tag{49}$$

With these closed-form expressions, we can solve the calibration problems exactly:

**Corollary G.2** (Advantage calibration for GM). *For a GM $M_\sigma(S)$ and target $\eta^\star > 0$, choosing $\sigma$ as:*

$$\sigma = \frac{\Delta_2}{2\Phi^{-1}\left(\frac{\eta^\star + 1}{2}\right)} \tag{50}$$

*ensures that adversary's advantage is upper bounded by $\eta^\star$.*

*Proof of Corollary G.2.* It is sufficient to ensure $(0, \eta^\star)$-DP. Plugging in $\varepsilon = 0$ and $\delta = \eta^\star$ into Eq. (48), we have:

$$\eta^\star = \Phi\left(\frac{\mu}{2}\right) - \Phi\left(-\frac{\mu}{2}\right) = 2\Phi\left(\frac{\mu}{2}\right) - 1, \tag{51}$$

from which we can derive $\mu = \frac{\Delta_2}{\sigma} = 2\Phi^{-1}\left(\frac{\eta^\star + 1}{2}\right)$ □

By solving Eq. (49) for $\alpha$, we also have an exact expression for calibrating to a given level of $\alpha^\star, \beta^\star$:

**Corollary G.3** (FPR/FNR calibration for GM). *For a Gaussian mechanism $M_\sigma(S)$, and target $\alpha^\star \geq 0$, $\beta^\star \geq 0$ such that $\alpha^\star + \beta^\star \leq 1$, choosing $\sigma$ as:*

$$\sigma = \frac{\Delta_2}{\Phi^{-1}(1 - \alpha^\star) - \Phi^{-1}(\beta^\star)} \tag{52}$$

*ensures that adversary's FNR and FPR rates are lower bounded by $\alpha^\star$ and $\beta^\star$, respectively.*

Note that using the exact expressions above to calibrate Gaussian mechanism offer only computational advantages compared the method in the main body. In terms of resulting noise scale $\sigma$, the results are the same as with generic PLRV-based calibration up to a numerical approximation error.

# H   Additional Experiments, Details, and Figures

## H.1   Computing Resources

We use a commodity machine with AMD Ryzen 5 2600 six-core CPU, 16GB of RAM, and an Nvidia GeForce RTX 4070 GPU with 16GB of VRAM to run our experiments. All experiments with deep learning take up to four hours to finish.

## H.2   Experimental Setup

In all our experimental results, the neighborhood relation $S \simeq S'$ is the add-remove relation, i.e., $S \simeq S'$ iff $|S \Delta S'| = 1$, which is the standard relation used by modern DP-SGD accountants. See more on implementation details related to the neighborhood relation in Appendix F.

**Text Sentiment Classification.**   We follow Yu et al. (2021) to finetune a GPT-2 (small) (Radford et al., 2019) using LoRA (Hu et al., 2021) with DP-SGD on the SST-2 sentiment classification task (Socher et al., 2013) from the GLUE benchmark (Wang et al., 2018). We use the Poisson subsampling probability $p \approx 0.004$ corresponding to expected batch size of 256, gradient clipping norm of $\Delta_2 = 1.0$, and finetune for three epochs with LoRA of dimension 4 and scaling factor of 32. We vary the noise multiplier $\sigma \in \{0.5715, 0.6072, 0.6366, 0.6945, 0.7498\}$ approximately corresponding to $\varepsilon \in \{3.95, 3.2, 2.7, 1.9, 1.45\}$, respectively, at $\delta = 10^{-5}$. We use the default training split of the SST-2 dataset containing 67,348 examples for finetuning, and the default validation split containing 872 examples as a test set.

**Image Classification.**   We follow Tramer and Boneh (2021) to train a convolutional neural network (Tramer and Boneh, 2021, Table 9, Appendix) over the ScatterNet features (Oyallon and Mallat, 2015) on the CIFAR-10 (Krizhevsky et al., 2009) image classification dataset. We use the Poisson subsampling probability of $p \approx 0.16$ corresponding to expected batch size of 8192, learning rate of 4, Nesterov momentum of 0.9, and gradient clipping norm of $\Delta_2 = 0.1$. We train for up to 100 epochs. We vary the gradient noise multiplier $\sigma/\Delta_2 \in \{4, 5, 6, 8, 10\}$, corresponding to $\varepsilon \in \{5, 3.86, 3.15, 2.31, 1.63\}$, respectively, at $\delta = 10^{-5}$. We use the default 50K/10K train/test split of CIFAR-10.

## H.3   Additional Experiments with Histogram Release

Histogram release is a simple but common usage of DP, appearing as a building block, e.g., in private query interfaces (Gaboardi et al., 2020). To evaluate attack-aware noise calibration for histogram release, we use the well-known ADULT dataset (Becker and Kohavi, 1996) comprising a small set of US Census data. We simulate the release of the histogram of the 'Education' attribute (with 16 distinct values, e.g., "High school", "Bachelor's", etc.) using the standard Gaussian mechanism with post-processing to ensure that the counts are positive integers. To measure utility, we use the $L_1$ distance (error) between the original histogram and the released private histogram.

Figure 7 shows the increase in utility if we calibrate the noise of the mechanism using the direct calibration algorithm to a given level of FPR $\alpha^\star$ and FNR $\beta^\star$ vs. standard calibration over 100 simulated releases with different random seeds. In certain cases, e.g., for $\alpha^\star = 0.1$ and $\beta^\star = 0.75$, our approach decreases the error by approx. $3\times$ from three erroneous counts on average to one.

### H.4 Software

We use the following key open-source software:

- PyTorch (Paszke et al., 2019) for implementing neural networks.
- huggingface (Wolf et al., 2019) suite of packages for training language models.
- opacus (Yousefpour et al., 2021) for training PyTorch neural networks with DP-SGD.
- dp-transformers (Wutschitz et al., 2022) for differentially private finetuning of language models.
- numpy (Harris et al., 2020), scipy (Virtanen et al., 2020), pandas (pandas development team, 2020), and jupyter (Kluyver et al., 2016) for numeric analyses.
- seaborn (Waskom, 2021) for visualizations.

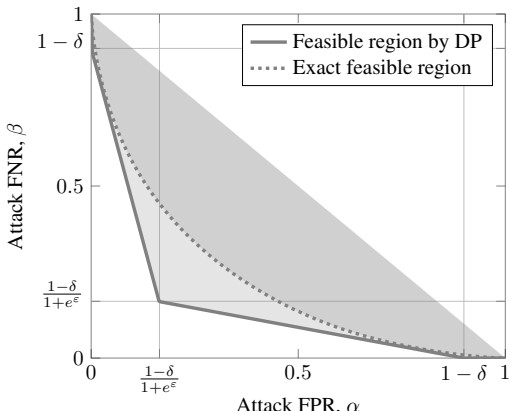

**Figure 5:** Trade-off curves of a Gaussian mechanism that satisfies $(\varepsilon, \delta)$-DP. Each curve shows a boundary of the feasible region (greyed out) of possible membership inference attack FPR ($\alpha$) and FNR ($\beta$) pairs. The solid curve shows the limit of the feasible region guaranteed by DP via Eq. (5), which is a conservative overestimate of attack success rates compared to the exact trade-off curve (dotted). The maximum advantage $\eta$ is achieved with FPR and FNR at the point closest to the origin.

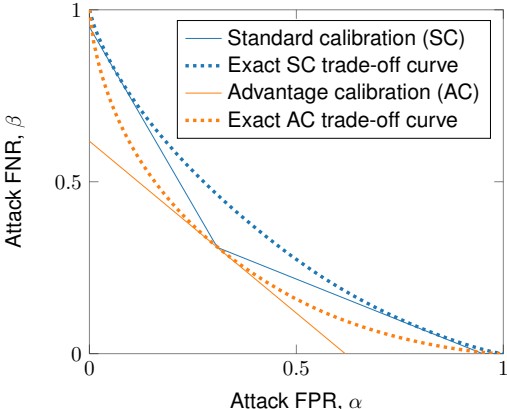

**Figure 6:** The increase in attack sensitivity due to calibration for advantage is less drastic for Gaussian mechanism than for a generic $(\varepsilon, \delta)$-DP mechanism.

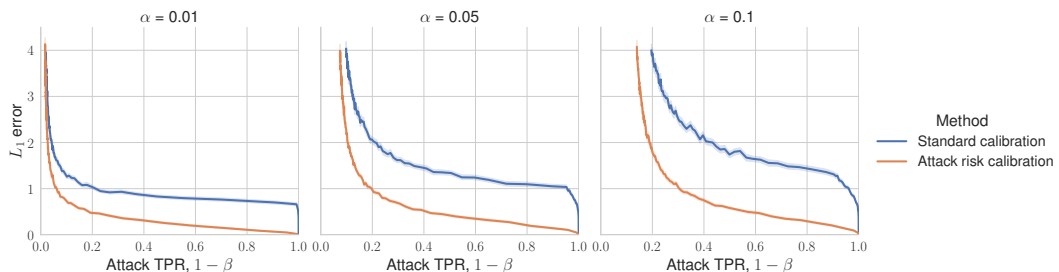

**Figure 7:** Direct calibration to attack FNR/FPR reduces average $L_1$ error in histogram release with Gaussian mechanism. The confidence bands are 95% CI over 100 simulated releases.

