# OpenReview forum: "Attack-Aware Noise Calibration for Differential Privacy"
_NeurIPS.cc/2024/Conference — NeurIPS 2024 poster_

### Official Review · Reviewer_tsqL · 2024-06-17

**Soundness:** 4
**Presentation:** 3
**Contribution:** 3
**Rating:** 5
**Confidence:** 4

**Summary:**

The paper proposes calibrating the noise in privacy mechanisms directly to MIA success metrics like advantage or TPR at low FPR instead of calibrating to a given $(\epsilon, \delta)$-bound and converting that to bounds on MIA success. The paper develops an algorithms for direct calibration for various MIA success metrics, most importantly TPR at low FPR. The algorithms use a discrete privacy loss random variable that is output by an existing privacy accountant. The paper compares the calibration methods by comparing the noise variances, and finds that calibrating directly significantly reduces the amount of noise required. The paper also does a small experiment showing that the reduction in noise translates to improved accuracy for machine learning.

**Strengths:**

The paper is mostly easy to read and understand, given the amount of theory. The case for calibrating directly to membership inference success metrics is made very well, and the algorithm for the calibration seems very practical.

**Weaknesses:**

Some parts of the proof of Theorem 3.4, the main theorem of the work, are not fully clear, and could be missing details needed to make them fully correct. I've collected these points in the Questions section. I think the issues can be fixed, and the theorem is likely true, but these should be addressed before the paper is accepted.

Figure 5 shows that the $\delta'$ corresponding to the attack TPR/FPR calibration is very large compared to the standard $\delta$. The standard $\delta$ is chosen to be small, since the mechanism that randomly outputs one individual's data is $(0, 1/n)$-DP with $n$ datapoints. I think the paper should discuss how this mechanism behaves with the TPR/FPR calibration. My quick calculation suggests that with the optimal attack for this mechanism, $FNR = (1 - 1/n) \cdot (1 - FPR)$, which would give a large FNR at small FPR, suggesting that the mechanism is very private, even though it obviously isn't. Requiring the FNR bound to be large to fix this could reduce the apparent advantage that the paper's results suggest, since the paper always considers smaller FNR bounds.

Minor points:
- Line 97: the domain and range of $\epsilon_\omega$ are the wrong way around.
- The neighbourhood relation is inconsistent: Section 2.1 describes substitute neighbourhood, while Section 2.2 describes add/remove neighbourhood.
- TPR does not necessarily need to be high for a MIA to be a relevant threat. For example, if TPR = 1% at FPR = 0.01%, the attack can very accurately identify the presence of a small number of individuals, which violates the privacy of those individuals.
- The results in Figure 1 could have uncertainty estimates.
- Restating the theorems before their proofs and having the proofs in the same order as the main text would make the proofs much easier to read.
- Line 649: $\phi$ outputs the probability that the sample came from $Q$.

**Questions:**

Issues with Theorem 3.4:
- In the reasoning around lines 673-675, why is it not possible that $\alpha$ just happens to be one of the $k+1$ possible values of Eq. (42)? On line 659, it is said that this reasoning should work for all $\alpha$. This case is considered explicitly later, but only after this line of reasoning is concluded.
- Doesn't the conclusion that there can be multiple ways values of $\gamma$ and $\tau$ that satisfy the constraint on line 695 imply that the choice of the optimal test $\phi_{\tau,\gamma}$ is not unique, though the FPR and FNR are unique?
- Why is the test that is found on lines 690-698 optimal?
- It is not clear whether Theorem 3.4 claims "for all $\tau$ and $\gamma$" or "for some $\tau$ and $\gamma$".

Minor points:
- I don't think Algorithm 1, line 2 works with $\alpha = 1$, the strict inequality is never satisfied. Though this should be easy to fix.
- Line 164: should this have $\alpha^\star$ instead of $\alpha$?
- Are $P$ and $Q$ the correct way around in Eq. (1)? They seem to be the other way around in Gopi et al. [2021].
- Should the support of $Y$ in line 666 be $y_1, \dotsc, y_{l+1}$ instead of $x_1, \dotsc, x_{l+1}$?
- Are the image classification results in Figure 1 comparable between standard and attack risk calibrations, since you had to use RDP due to data augmentations with the standard accountant?
- What type of subsampling (Poisson, sampling with/without replacement) and which neighbourhood relation did you use in the DP-SGD experiments? The fixed batch sizes suggest sampling without replacement, in which case the issues that Lebeda et al. (2024) have recently raised with privacy accounting for sampling without replacement might affect your results.

Reference:
- Lebeda et al. "Avoiding Pitfalls for Privacy Accounting of Subsampled Mechanisms under Composition" (2024) arXiv:2405.20769

**Limitations:**

The paper discusses limitations adequately.

---

> ### Author Rebuttal · Authors · 2024-08-07
>
> We would like to thank you for such a detailed reading of our work, and especially the proof! We appreciate it.
>
> ## Theorem 3.4
>
> > **Q1.** In the reasoning around lines 673-675, why is it not possible that $\alpha$ just happens to be one of the possible values of Eq. (42)? On line 659, it is said that this reasoning should work for all $\alpha$. This case is considered explicitly later, but only after this line of reasoning is concluded.
>
> We agree that this line of reasoning appeared too early in the proof. We will move it to the part of the proof where we only consider $\alpha$ that are not one of the possible values of the reverse CDF of $X$ in Eq. (42).
>
> > **Q2.** Doesn't the conclusion that there can be multiple ways values of $\gamma$ and $\tau$ that satisfy the constraint on line 695 imply that the choice of the optimal test is not unique, though the FPR and FNR are unique?
>
> Not necessarily! What we are saying in line 695 is that the optimal test, which is unique, can have two different _parameterizations:_ $(\tau_1 = x_t, \gamma_1 = 0)$ and $(\tau_2 = x_{t+1}, \gamma_2 = 1)$. That these two parameterizations yield the same test follows by observing that $\phi^*_{\tau_1, \gamma_1}(o)$ outputs 1 if $\log \frac{Q(o)}{P(o)} > x_t$ and 0 otherwise, while $\phi^*_{\tau_2, \gamma_2}(o)$ outputs 1 if $\log \frac{Q(o)}{P(o)} \geq x_{t+1}$ and 0 otherwise. Since the test statistic $\log \frac{Q(o)}{P(o)}$ lives on a discrete grid and cannot take on values between $x_t$ and $x_{t+1}$, it follows that $\phi^*_{\tau_1, \gamma_1}$ and $ \phi^*_{\tau_2, \gamma_2}$ will classify each observation $o$ identically.
>
> > **Q3.** Why is the test that is found on lines 690-698 optimal?
>
> Our response to Question 2 should address this concern as well. Lines 690-698 do NOT identify two distinct tests and choose one as the optimal without proof. Instead, lines 690-698 identify two different parameterizations for the unique and optimal test. One is free to choose either parameterization (or any other!) for implementation. In our implementation, we choose $(\tau_1 = x_t, \gamma_1 = 0)$. We will make this distinction between different parameterizations and different tests clear in the final version of the proof (see next).
>
> > **Q4.** It is not clear if Theorem 3.4 claims "for all" or "for some" $\gamma$ and $\tau$.
>
> Theorem 3.4 indeed holds **for all** $\tau \in \mathbb{R}$ and $\gamma \in [0,1]$, and we will make this clear by specifying the “for all” quantifier. To clarify this in the proof, in each of the three studied regions of $\alpha$, we will specify (i) the optimal test and (ii) all possible parameterizations $(\tau, \gamma)$ for the optimal test, as described next.
>
> In the case when $\alpha = 1$, we have only one possible parameterization for the optimal test: $(\tau = -\infty, \gamma = 0)$. When $\alpha$ is one of the $k+1$ values in Eq. 42, we have the two parameterizations in Question 2 along with a continuum $\{ (\tau, \gamma) \mid \tau \in (x_t, x_{t+1}), \gamma \in [0,1] \}$ if $t<k$, and $\{ (\tau, \gamma) \mid \tau \in (x_k, \infty), \gamma \in [0,1]\}$ if $t = k$. Finally, when $\alpha$ does not fall into any of the remaining categories, we have one unique parameterization $(\tau = x_t, \gamma = \text{Eq (49)})$ for all $t$.
>
> This covers all $\tau \in \mathbb{R}, \gamma \in [0,1]$, and shows that Theorem 3.4 claims holds "for all'' parameter values.
>
> ## Catastrophic Failures
>
> > Figure 5 shows that the $\delta'$ corresponding to the attack TPR/FPR calibration is very large compared to the standard $\delta$. The standard $\delta$ is chosen to be small, as the mechanism that randomly outputs one individual's data is $(0, 1/n)$-DP with $n$ datapoints. I think the paper should discuss how this mechanism behaves with the TPR/FPR calibration.
>
> This is a great point that we thought about a lot, although a detailed discussion didn't make the final cut in this submission (it will be added to the revised version). The reviewer correctly notes that the standard convention of computing $\varepsilon$ for $\delta \ll \frac{1}{n}$ is based on the _existence_ of an $(\varepsilon, \delta)$-DP mechanism allowing catastrophic failures with probability $\delta$. Given our calibration in Figure 5 allows for $\delta > \frac{1}{n}$, there exists a mechanism achieving our target TPR/FPR with a high probability of catastrophic failure, and the reviewer provided an example of such mechanism.
> However, we introduce our calibration approach in the context of choosing a noise scale $\omega$ for a _given parameterized mechanism_ $M_\omega$. Most if not all practical mechanisms, including those we investigated (DP-SGD), do not admit catastrophic failures (see, e.g., this [post](https://differentialprivacy.org/flavoursofdelta/)). We argue that the _existence_ of a mechanism achieving our TPR/FPR calibration with a high probability of catastrophic failure is irrelevant when calibrating the noise parameter $\omega$ of a _specific mechanism_ that we know does not admit failures. Consequently, our methods should only be used for mechanisms without catastrophic failures. We will clarify this in the final version by adding the following paragraph in the Concluding Remarks:
>
> “Our calibration algorithms are supposed to be used with mechanisms that do not admit catastrophic failures, i.e., those that ensure attack TPR of zero when FPR is also zero, or, equivalently, mechanisms whose trade-off curve is such that $T(M(S), M(S’))(0) = 1$ and $T(M(S), M(S’))(1) = 0$ for all $S \simeq S’$. Although practical mechanisms such as DP-SGD have this property, one can in principle construct mechanisms which do admit catastrophic failures (e.g., a “name-and-shame” mechanism which outputs one of the records in the clear [see, e.g., [Aerni et al. 2024](https://arxiv.org/abs/2404.17399)]). In such pathological cases, standard calibration should be used to ensure $\delta \ll \frac{1}{n}$.”
>
> ---
> We respond to the minor points in the next comment.

---

> ### Author Response · Authors · 2024-08-07
> **Additional response regarding the minor points**
>
> > The neighbourhood relation is inconsistent: Section 2.1 describes substitute neighbourhood, while Section 2.2 describes add/remove neighbourhood.
>
> Thank you for pointing this out! Our analyses are independent of the choice of the neighborhood relation, and our experiments are done with the add-remove relation. We will remove the substitution relation in Section 2.1 paragraph “Setup and notation”, mention that we use the add-remove relation in our experiments in Section 2.1, paragraph “DP-SGD”. Moreover, in Section 2.2 we will emphasize that our results are not tied to the add/remove relation.
>
> > TPR does not necessarily need to be high for a MIA to be a relevant threat. For example, if TPR = 1% at FPR = 0.01%, the attack can very accurately identify the presence of a small number of individuals, which violates the privacy of those individuals.
>
> This is a great point! Ultimately, defining acceptable thresholds on TPR/FPR, as we mention in the Concluding Remarks, is an open problem. We note that acceptable thresholds can be made in a context and application specific manner. For example, from the computer security literature (see, e.g., [Wang, 2018](https://arxiv.org/abs/1802.05409)), we know that if the prior probability of membership is lower than the attack’s FPR, then the majority of the attack's positive predictions will be incorrect, even if TPR is 100% (Wang 2018, p.2; formally, prior-aware positive predictive value, ppv = P(member | attack predicts 'member') can be low even if TPR = P(attack predicts 'member' | member) is high, as it depends on the base rate P(member) and is more significantly influenced by FPR than TPR). Our framework fits nicely with this literature. If in a particular context we have a guess for the adversary’s prior probability, this can be used to inform an acceptable TPR/FPR thresholds.
>
> We will expand the short discussion in the Concluding Remarks on this limitation as follows:
>
> “We leave open the question on how to choose the target FPR $\alpha^\star$, e.g., whether standard significance levels in sciences such as $\alpha^\star = 0.05$ are compatible with data protection regulation, **as well as what are the acceptable attack TPR levels for a given FPR**. Future work is needed to develop concrete guidance on the choice of target FPR **and TPR** informed by legal and practical constraints.”
>
> > The results in Figure 1 could have uncertainty estimates.
>
> Thank you for the suggestion. In the new experimental demonstration on private histogram release, described in the [general response](https://openreview.net/forum?id=hOcsUrOY0D&noteId=4H6Ec2IhLz), we added uncertainty regions for utility measurements done over 100 mechanism runs with different random seeds.
>
> > Restating the theorems before their proofs and having the proofs in the same order as the main text would make the proofs much easier to read.
>
> We will implement this in the final version!
>
> ---
>
> > Are $P$ and $Q$ the correct way around in Eq. (1)? They seem to be the other way around in Gopi et al. [2021].
>
> $P$ and $Q$ are the correct way in Gopi et al. We will fix this in the final version.
>
> > Are the image classification results in Figure 1 comparable between standard and attack risk calibrations, since you had to use RDP due to data augmentations with the standard accountant?
>
> Thank you for raising this point. We indeed agree that this comparison is somewhat unfair. We now reanalyzed the training pipeline from Tramer & Boneh, 2021 using Doroshenko et al. accounting so that the methods are exactly comparable. We provide the updated figure in the PDF attached in the [general response](https://openreview.net/forum?id=hOcsUrOY0D&noteId=4H6Ec2IhLz). Indeed, the TPR values obtained with standard calibration and tight accounting are somewhat lower than with RDP accounting, especially in the small $\alpha$ regime, but the general trend remains: attack-aware calibration significantly increases task accuracy at the same risk level.
>
> > What type of subsampling and which neighbourhood relation did you use in the DP-SGD experiments?
>
> We use the add-remove relationship in our experiments, as it is standard with modern accountants. We use Poisson sampling. The batch size in the experimental details is supposed to mean the _expected batch size._ To avoid confusion, we will clearly specify the neighborhood relation and write in the format “subsampling rate $p = 0.003801$ (corresponding to the expected batch size of 256)”.
>
> > Line 97: the domain and range of epsilon_omega are the wrong way around. Line 164: should this have $\alpha^\star$  instead of $\alpha$? Should the support of $Y$ in line 666 be $\\{y_1, …, y_{l+1}\\}$ instead of $\\{x_1, …, x_{l+1}\\}$? Line 649: $\phi$ outputs the probability that the sample came from $Q$. I don't think Algorithm 1, line 2 works with $\alpha = 1$, the strict inequality is never satisfied. Though this should be easy to fix.
>
> That’s right, thank you so much for spotting these! We will fix them.

---

> ### Comment · Reviewer_tsqL · 2024-08-08
>
> Thank you for the comprehensive response. You addressed most of my concerns, most importantly those regarding the proof of Theorem 3.4.
>
> Regarding my point about catastrophic failures, it seems that I didn't make my concern quite clear in the initial review. You are correct in saying that many mechanism do not actually allow the catastrophic privacy failure, so it may not be necessary to set a conservative TPR@FPR bound to account for them.
>
> However, the same reasoning can be used to argue that we do not need $\delta \ll \frac{1}{n}$. As a result, the utility increase of TPR@FPR calibration over $(\epsilon, \delta)$-calibration seems to come from using this argument for one definition, but not the other. Reading the paper again, I noticed that you have alluded to this in the caption of Figure 5, but this point is important enough to discuss in the main text. Currently, the Abstract and Introduction give the impression that the utility benefit is due to some intrinsic difference between $(\epsilon, \delta$) and TPR@FPR bounds, and not simply a consequence of what values are typically considered acceptable for the bounds.
>
> On the other hand, I recognise that calibrating to attack risk is useful, and your method for this calibration is much easier to use than first finding the optimal $(\epsilon, \delta)$-bound corresponding to the attack risk, and then calibrating the mechanism. As a result, the paper makes a valuable contribution, and I'm increasing my score accordingly.

---

> > ### Author Response · Authors · 2024-08-08
> >
> > > You are correct in saying that many mechanism do not actually allow the catastrophic privacy failure, so it may not be necessary to set a conservative TPR@FPR bound to account for them … However, the same reasoning can be used to argue that we do not need $\delta \ll \frac{1}{n}$ …  Currently, the Abstract and Introduction give the impression that the utility benefit is due to some intrinsic difference between $(\varepsilon, \delta)$ and TPR@FPR bounds, and not simply a consequence of what values are typically considered acceptable for the bounds.
> >
> > Thank you for raising this point. Just to clarify, as long as the TPR@FPR bounds are computed using _the privacy curve $\varepsilon(\delta)$ (using the approaches in Appendix A) / PLRVs $(X, Y)$ (using Algorithm 1)_ and not based on a single $(\varepsilon, \delta)$-point (using Eq. 5), then indeed there is no intrinsic difference. There **is**, however, an intrinsic difference in calibrating to a target TPR@FPR using Algorithm 1 vs. calibrating to a target TPR@FPR with _a fixed delta_ (standard calibration), i.e., the difference between the two curves in Figure 4.
> >
> > We appreciate your time and willingness to revise your review, and we are happy to answer any other questions you may have. Your input has been very valuable!

---

### Official Review · Reviewer_pcXQ · 2024-07-10

**Soundness:** 3
**Presentation:** 3
**Contribution:** 3
**Rating:** 6
**Confidence:** 3

**Summary:**

This paper proposes new methods for calibrating noise in differentially private learning to achieve a given level of operational privacy risk, specifically focusing on the advantage and FNR/FPR of membership inference attacks. The methods reduce the noise scale compared to the standard two-step procedure (first converting to a privacy budget, then converting to a privacy assessment).

**Strengths:**

1. The paper addresses an important practical problem in the privacy regime with significant theoretical contributions.

2. It is well-organized and easy to follow.

**Weaknesses:**

1. I suggest the authors include more downstream tasks and utility metrics to further demonstrate the effectiveness of the theoretical results.

2. High-level intuitions for different variables in Theorem 3.4, such as $\alpha(\tau, \gamma)$ and $\beta(\tau, \gamma)$, would be beneficial.

**Questions:**

1. Will choosing a discrete-valued dominating pair of the mechanism be a sub-optimal choice for advantage calibration?

2. How is the discretized PLRV typically obtained for a general mechanism?

---

> ### Author Rebuttal · Authors · 2024-08-07
>
> Thank you for the review and suggestions!
>
> **Weaknesses**
>
> > I suggest the authors include more downstream tasks and utility metrics to further demonstrate the effectiveness of the theoretical results.
>
> We have added a new experiment on a common use case of DP – private histogram release – which shows that attack-aware noise calibration also enables to significantly reduce the error when privately releasing histograms. See the [general response](https://openreview.net/forum?id=hOcsUrOY0D&noteId=4H6Ec2IhLz) and the attached PDF for details.
>
> > High-level intuitions for different variables in Theorem 3.4, such as $\alpha(\tau, \gamma)$ and
> $\beta(\tau, \gamma)$, would be beneficial.
>
> We will add the following lines after Theorem 3.4:
>
> “The proof for Eq. (12) works by using the Neyman-Pearson lemma to explicitly construct the optimal most powerful attack at level $\alpha$. We parameterize the optimal attack in terms of intermediate parameters $\tau$ and $\gamma$, where $\tau$ is the threshold for the Neyman-Pearson test statistic and $\gamma$ is the probability of guessing in case the test statistic exactly equals the threshold.”
>
>
> **Questions**
> > Will choosing a discrete-valued dominating pair of the mechanism be a sub-optimal choice for advantage calibration?
>
> Any dominating pair, whether discrete or continuous, suffices for advantage calibration. Moreover, the results of Doroshenko et al. 2022 imply that a carefully-crafted discrete-valued dominating pair can provide arbitrarily tight privacy bounds to a continuous mechanism. Since we use the algorithm proposed by Doroshenko et al., the answer to this question is no: a discrete-valued dominating pair is not sub-optimal for advantage calibration.
>
> > How is the discretized PLRV typically obtained for a general mechanism?
>
> In Appendix E, we detail the technique from Doroshenko et al. for constructing PLRVs for a general mechanism given its privacy curve. The technique discretizes the privacy profile curve, and builds pmfs of the dominating pair $P$, $Q$ using the discretized values. The distributions of the PLRVs can then computed using the pmfs of $P$, $Q$.

---

> > ### Comment · Reviewer_pcXQ · 2024-08-13
> >
> > Thanks for the rebuttal! I don't have further questions.

---

### Official Review · Reviewer_Bdkd · 2024-07-13

**Soundness:** 4
**Presentation:** 2
**Contribution:** 4
**Rating:** 7
**Confidence:** 2

**Summary:**

This paper introduces a new method for improving  the utility of privacy-preserving machine learning without sacrificing privacy protection. The authors develop efficient algorithms to calculate the trade-off curve between attack FPR and FNR using f-differential privacy (f-DP). They then show how to use this information to fine-tune the amount of noise added, allowing for precise control over privacy risks. This approach offers a more refined way to balance data usefulness and privacy in machine learning, addressing a key challenge in the field.

**Strengths:**

The authors show that their direct calibration methods can significantly reduce the required noise scale compared to the standard approach, leading to an increase in utility (e.g., 18 percentage points higher accuracy) for the same level of privacy risk. They also demonstrate that calibrating for attack advantage (attack accuracy) can increase attack power in the low FPR regime, and show that calibrating for a desired FPR and FNR level mitigates this issue. By this method, the noise of DPSGD can be reduced tighter once we are aware of the privacy risk.

**Weaknesses:**

The notation in this paper is heavy, could you provide notation tables?


Could you provide a more detailed algorithm in the appendix about how to use $\epsilon$ , $\delta$, and q to generate Advantage Calibration’s Xω , Yω.  Its quite hard to follow in Section 3’s demonstration. I think authors need an algorithm to demonstrate how they did the experiment from the supplement's code.

I am willing to increase my score if the above questions can be answered.

**Questions:**

see in weaknesses

**Limitations:**

see in weaknesses

---

> ### Author Rebuttal · Authors · 2024-08-07
>
> Thank you for your review and suggestions. We address some of points raised directly below, but they are also partially covered in our general response, to which we refer when relevant.
>
> > The notation in this paper is heavy, could you provide notation tables?
>
> We agree. See the [general response](https://openreview.net/forum?id=hOcsUrOY0D&noteId=4H6Ec2IhLz) for the notation table.
>
> > Could you provide a more detailed algorithm in the appendix about how to use $\epsilon$, $\delta$ , and $q$ to generate Advantage Calibration’s $X_\omega$ , $Y_\omega$. It’s quite hard to follow in Section 3’s demonstration. I think authors need an algorithm to demonstrate how they did the experiment from the supplement's code.
>
> In Section 3, following Theorem 3.4, we point to Appendix E for how we construct $X_\omega$, $Y_\omega$ for a general mechanism given its privacy curve. We will clarify in L226 as follows:
>
> “Given a method for obtaining valid PLRVs $X_\omega$ , $Y_\omega$ for any $\omega$, **such as the one provided by Doroshenko et al. 2022 (see Appendix E),** …”
>
> We want to emphasize to the reviewer that Appendix E is a summary of the algorithm proposed by Doroshenko et al. 2022, which we used in all of our experiments. Moreover, this algorithm uses the entire privacy curve and mechanism-specific functions such as $q$ to construct $X_\omega$, $Y_\omega$, not just a single $(\epsilon, \delta)$ pair.
>
> Additionally, we provide a more detailed description of the steps involved in the calibration algorithms in the [general response](https://openreview.net/forum?id=hOcsUrOY0D&noteId=4H6Ec2IhLz).

---

### Official Review · Reviewer_JyJW · 2024-07-28

**Soundness:** 3
**Presentation:** 3
**Contribution:** 3
**Rating:** 5
**Confidence:** 3

**Summary:**

Differential privacy (DP) mitigates privacy risks in machine learning by adding noise during training, balancing privacy and utility. Traditionally, the noise scale is set using a privacy budget parameter ε, which is then translated to attack risk. This two-step method often results in conservative risk assessments and reduced utility. The proposed approach directly calibrates noise to a desired attack risk, bypassing ε, thus decreasing the noise scale and enhancing utility while maintaining privacy. Empirical evidence shows that this method improves model accuracy for the same privacy level, offering a practical way to enhance privacy-preserving machine learning.

**Strengths:**

1. The proposed framework successfully addresses technical challenges, providing compelling insights.
2. The experimental results are robust and strongly support the framework's effectiveness.

**Weaknesses:**

1.	The paper includes numerous definitions and symbols, which can be confusing for readers. Creating a table to summarize these terms and explain their meanings would greatly enhance clarity and help readers follow along more easily.
2.	In Section 2, the problem statement is not clearly articulated, making it difficult to discern the main goal and the specific problem being addressed. Highlighting the main goal and explicitly defining the problem would make this section more readable and comprehensible.

**Questions:**

1.	In the experiment section, how about the results of other low FPR regimes?

**Limitations:**

1. In Section 3.1, please include inference for ensuring advantage calibration in the appendix.
2. The two figures at the top of page 7 lack figure names. It appears they should be labeled as Figure 2.

---

> ### Author Rebuttal · Authors · 2024-08-07
>
> Thank you for the review and the suggestions! We respond to the comments and questions next. Note that we also address some of them in the general response, and we refer to it when relevant.
>
> > The paper includes numerous definitions and symbols, which can be confusing for readers. Creating a table to summarize these terms and explain their meanings would greatly enhance clarity and help readers follow along more easily.
>
> See the [general response](https://openreview.net/forum?id=hOcsUrOY0D&noteId=4H6Ec2IhLz) for the notation table.
>
> > In Section 2, the problem statement is not clearly articulated, making it difficult to discern the main goal and the specific problem being addressed. Highlighting the main goal and explicitly defining the problem would make this section more readable and comprehensible.
>
> The problem statement appears in the discussion of Section 2.3. We will highlight this more clearly in the final version, and change the title of Section 2.3 to “Our Objective: Attack-Aware Noise Calibration” to clearly signal that the problem statement is there. To be clear, our problem is to solve the calibration problem $\min \omega \text{ s.t. } risk_\omega \leq threshold$, in particular, by providing a method to efficiently compute $risk_\omega$ for standard notions of operational attack risk in DP.
>
> **Questions**
>
> > In the experiment section, how about the results of other low FPR regimes?
>
> See the attached PDF in the [general response](https://openreview.net/forum?id=hOcsUrOY0D&noteId=4H6Ec2IhLz) for the trade-off curves for all five models in the language modeling experiment. Using these curves, one can glean the behavior in terms of FPR/FNR trade-off for different levels of noise scale/task accuracy and for all $\alpha \in [0, 1]$ as opposed to $\alpha \in \{{0.01, 0.05, 0.1\}}$. We will add these plots in the Appendix in the final version.
>
> **Limitations**
>
> > In Section 3.1, please include reference for ensuring advantage calibration in the appendix.
>
> Eq. 13 in Section 3.1 shows (1) the optimization problem corresponding to advantage calibration and (2) how we compute advantage given dominating PLRVs $(X_\omega, Y_\omega)$. Moreover, in Appendix E we explain how to obtain $X_\omega, Y_\omega$ from an arbitrary mechanism given its privacy curve, and in Section 2.3 we explain how we solve an optimization problem in the form of Eq 13. In the final version, we will consolidate all this information into one algorithm in the Appendix, as showed in the [general response](https://openreview.net/forum?id=hOcsUrOY0D&noteId=4H6Ec2IhLz).
>
> > The two figures at the top of page 7 lack figure names. It appears they should be labeled as Figure 2.
>
> Thank you for spotting this issue! Indeed, this Figure is missing a caption. We will fix this in the final version.

---

> > ### Author Response · Authors · 2024-08-12
> >
> > As the author-reviewer period is coming to end, please let us know if we have addressed your concerns and if we can clarify anything else.

---

### Author Rebuttal · Authors · 2024-08-07

We would like to thank all the reviewers for their time and feedback. We are glad the reviews found that our framework addresses an important technical problem (pcXQ), provides significant theoretical contributions (pcXQ), and compelling insights (JyJW). We are also glad that the reviews appreciated the practicality of our algorithms (tsqL), the increased utility that our approach enables (Bdkd), the robustness of the experimental results (JyJW), as well as pointed out that the paper is easy to follow (pcXQ, tsqL) and builds a strong case for attack-aware noise calibration (tsqL).

We noticed some general trends in the reviewers' suggestions, addressed next.

**Notation table.** Reviewers JyJW and Bdkd remarked that the paper is notation heavy, and asked for a table summarizing the different terms and their meanings. We agree and will add the notation table in the final version (see below).

**Algorithms for standard and direct advantage calibration.** Reviewers JyJW and Bdkd both asked for a single algorithm detailing how we ensure advantage calibration. We acknowledge that the exact steps for calibration are spread across Sections 2 and 3, and we will consolidate them in the final version. Specifically, we will detail the exact steps as follows:

_Standard calibration._ Inputs: privacy parameters $\eta^\star, \delta^\star < \frac{1}{n}$, privacy profile $\varepsilon_\omega(\delta)$.

1. Find $\varepsilon^\star$ which ensures $\eta = \eta^\star$ for a given $\delta^\star$, i.e., solve Eq. 8 for $\varepsilon$ for fixed $\delta = \delta^\star$ and $\eta = \eta^\star$.
2. Find noise parameter $\omega$ that ensures $(\varepsilon^\star, \delta^\star)$-DP using binary search as described in Sec. 2.3:

	$\min_\omega \text{ s.t. } \varepsilon_\omega(\delta^\star) \geq \varepsilon^\star$.

	This step is exactly Eq. 2.

_Direct advantage calibration (ours)._ Inputs: privacy parameter $\eta^\star$, PLRVs $X_\omega, Y_\omega$.

Find noise parameter $\omega$ that ensures $\eta^\star$ advantage using binary search as described in Sec. 2.3, and using Eq. 11 in Theorem 3.4 to instantiate the $risk_\omega = \eta_\omega = P[Y_\omega > 0] - P[X_\omega > 0]$ function:

$\min_\omega \text{ s.t. } P[Y_\omega > 0] - P[X_\omega > 0] \leq \eta^\star.$

This step is exactly Eq. 13.

**Additional plots and experiments.** Moreover, we attach a PDF with additional results:

1. Trade-off curves for the language modeling experiments, showing a more complete picture of attainable $(\alpha, \beta)$ values in addition to $\alpha \in \{{0.01, 0.05, 0.1\}}$ in the submission (following a question by JyJW).

2. A version of Figure 1 for the image classification experiments with a tight reanalysis of the method by Tramer and Boneh, 2021, using the Doroshenko et al. accountant instead of the RDP accountant to ensure fair comparisons (following a comment by tsqL).

3. Following a suggestion by pcXQ, we added a new experimental setting in which we use attack-aware noise calibration for releasing a single _differentially private histogram_. This is a simple but common usage of DP, appearing as a building block, e.g., in private query interfaces. To make it concrete, we use the well-known ADULT dataset comprising a small set of US Census data, and simulate the release of the histogram of the 'Education' attribute (with 16 distinct values, e.g., “High school”, “Bachelor’s”, etc.). To measure utility, we use the $L_1$ distance (error) between the original histogram and the released private histogram. The plot shows the increase in utility if we calibrate the noise of the Gaussian mechanism (with post-processing to ensure the counts are positive integers) using the direct calibration algorithm to a given level of FPR $\alpha^\star$ and FNR $\beta^\star$ vs. standard calibration over 100 simulated releases with different random seeds. In certain cases, e.g., for $\alpha^\star = 0.1$ and $\beta^\star = 0.75$, our approach decreases the $L_1$ error by 3x from three erroneous counts on average to one.

We address the other comments in individual responses.

---

| Symbol | Description | Reference |
|-|-|-|
| $z \in \mathbb{D}$ | Data record |       	|
| $S \in 2^{\mathbb{D}}$ | Dataset of records   	|
| $S \simeq S'$ | Adjacency relation of neighboring datasets that differ by one record |       	|
| $M_\omega: 2^{\mathbb{D}} \rightarrow \Theta$  | Privacy-preserving mechanism                                     	|       	|
| $\omega \in \Omega$                        	| Noise parameter of a given mechanism $M(S)$                      	|       	|
| $D_\gamma(M(S) \| M(S')) \geq 0,\ \gamma \geq 0$         	| Hockeystick divergence                                           	| Eq. 1 	|
| $\varepsilon \in (0, \infty), \delta \in [0, 1]$	| Privacy parameters in differential privacy                       	| Def 2.1   |
| $\varepsilon: \mathbb{R} \rightarrow [0, 1]$ | Privacy profile curve                                            	| Def 2.2   |
| $\phi: \Theta \rightarrow [0, 1]$ | Membership inference hypothesis test                    	|       	|
| $\alpha_\phi \in [0, 1]$  | False positive rate (FPR) of attack $\phi(\theta)$               	|       	|
| $\beta_\phi \in [0, 1]$  | False negative rate (FNR) of attack $\phi(\theta)$               	|       	|
| $\eta \in [0, 1]$   | Maximal advantage across attacks against mechanism $M(S)$        	| Eq. 7 	|
| $T(M(S), M(S')): [0, 1] \rightarrow [0, 1]$ | Trade-off curve between FPR and FNR of optimal attacks           	| Def. 2.3  |
| $f: [0, 1] \rightarrow [0, 1]$  | A lower bound on the trade-off curve for all neighboring datasets   | Def. 2.4  |
| $P, Q$  | A dominating pair of distributions for a given mechanism $M(S)$  	| Def 3.1   |
| $X, Y$  | Privacy loss random variables for a given dominating pair $P, Q$ 	| Def 3.2   |

---

### Author Response · Authors · 2024-08-14

We would like to sincerely thank all reviewers for their constructive and actionable feedback, and for responsiveness during the discussion period. We believe the changes following the feedback, which we detailed in the individual responses and in the [general response](https://openreview.net/forum?id=hOcsUrOY0D&noteId=4H6Ec2IhLz), will significantly strengthen our paper.

---

### Decision · Program_Chairs · 2024-09-25

**Decision:**

Accept (poster)

**Comment:**

The paper presents a novel one-step for procedure for determining the noise level needed to prevent inference attacks on private data, in contrast to the usual two-step procedure that is routed through a DP guarantee. All reviewers agreed that the paper should be accepted.